# Chord: an ensemble machine learning algorithm to identify doublets in single-cell RNA sequencing data

Ke-Xu Xiong[1,2,9], Han-Lin Zhou [2,3,4,5,6,9,10✉], Cong Lin[2,4,5,7], Jian-Hua Yin[2,4,5,7], Karsten Kristiansen[2,6], Huan-Ming Yang[2,8] & Gui-Bo Li [2,3,4,5,7,10✉]

High-throughput single-cell RNA sequencing (scRNA-seq) is a popular method, but it is accompanied by doublet rate problems that disturb the downstream analysis. Several computational approaches have been developed to detect doublets. However, most of these methods may yield satisfactory performance in some datasets but lack stability in others; thus, it is difficult to regard a single method as the gold standard which can be applied to all types of scenarios. It is a difficult and time-consuming task for researchers to choose the most appropriate software. We here propose Chord which implements a machine learning algorithm that integrates multiple doublet detection methods to address these issues. Chord had higher accuracy and stability than the individual approaches on different datasets containing real and synthetic data. Moreover, Chord was designed with a modular architecture port, which has high flexibility and adaptability to the incorporation of any new tools. Chord is a general solution to the doublet detection problem.

[1] College of Life Sciences, University of Chinese Academy of Sciences, Beijing 100049, China. [2] BGI-Shenzhen, Shenzhen 518083, China. [3] BGI College & Henan Institute of Medical and Pharmaceutical Science, Zhengzhou University, Zhengzhou, China. [4] BGI-Henan, BGI-Shenzhen, Xinxiang 453000, China. [5] Guangdong Provincial Key Laboratory of Human Disease Genomics, Shenzhen Key Laboratory of Genomics, BGI-Shenzhen, Shenzhen 518083, China. [6] Laboratory of Genomics and Molecular Biomedicine, Department of Biology, University of Copenhagen, Copenhagen DK-2100, Denmark. [7] Shenzhen Key Laboratory of Single-Cell Omics, BGI-Shenzhen, Shenzhen 518083, China. [8] James D. Watson Institute of Genome Science, 310008 Hangzhou, China. [9] These authors contributed equally: Ke-Xu Xiong, Han-Lin Zhou. [10] These authors jointly supervised this work: Han-Lin Zhou, Gui-Bo Li. ✉email: zhouhanlin@genomics.cn; liguibo@genomics.cn

Recently, the development of high-throughput single-cell RNA sequencing (scRNA-seq) has provided convenience for dissecting the cellular heterogeneity of tissues[1]. In contrast to bulk RNA sequencing, profiling transcriptomes at the single-cell resolution has enabled researchers to recognise the molecular characteristics of all cell types at one time and acquire a better insight into physiology, biological development, and disease[2]. Among the current state-of-the-art technologies of high-throughput scRNA-seq, droplet-based technologies are currently commonly employed as an unbiased solution of single-cell transcriptomics[3]. However, these microfluidic methods often encounter the problem of doublets, where one droplet may contain two or more cells with the same barcode during the distribution step of isolating single cells. Then the doublets are counted as a single cell in the data forming technical artefacts[4]. According to the composition of doublets, doublets can be divided into two major classes: homotypic doublets, which originate from the same cell type, and heterotypic doublets, which arise from distinct transcriptional cells generating an artificial hybrid transcriptome[4,5]. Compared to homotypic doublets, heterotypic doublets are considered to have more impact on downstream analyses, including dimensionality reduction, cell clustering, differential expression, and cell developmental trajectories[6,7].

To reduce the number of doublets in experiments, decreasing the concentration of loaded cells is an effective control measure. However, this approach also reduces the number of captured cells and dramatically increases the cost per sample[6,8]. Several existing experimental techniques can be applied to identify doublets instead of avoiding doublets, such as the cell hashing method using oligo-tagged antibodies as an orthogonal information[9], MULTI-seq using lipid-tagged indices[10], and demuxlet using natural genetic variations[11]. However, there are inherent limitations to these experimental techniques. First, since these methods require special experimental operations and additional costs, so they are not helpful for the existing scRNA-seq data. Second, these techniques only experimentally label doublets from different samples but ignore the kind of doublet generated by cells from the same sample or individual. Therefore several computational approaches have been developed to detect doublets in common scRNA-seq data, including data already generated[7]. A benchmarking study has shown that the performance of these computational methods varies greatly, even the top-performing methods with the noticeable differences[7], so there is still a larger challenge in terms of sub-optimal accuracy of each method. In addition, because of the unique characteristics brought by these specific mathematical algorithms and their applicability to different scenarios, no method can be considered as the gold standard for each scenario; Thus, it is a challenging and time-consuming task for researchers to choose suitable software for their specific research.

To address these unmet needs, we propose Chord, which implements an ensemble algorithm that aggregates the results from multiple representative methods to identify doublets accurately. The ensemble algorithm is a widely used technique of machine learning[12] that can boost the accuracy of somatic mutation detection[13] and culprit lesion identification[14]. Compared to the individual methods, Chord was demonstrated an improved accuracy and stability in doublet detection across different datasets of real and synthetic data. Moreover, Chord was designed with a modular architecture port that is highly flexible and adaptable to incorporate new tools.

## Results

### The Chord workflow for accurate and robust ensemble algorithm-based doublet detection.
The computational approaches to detect doublets in scRNA-seq data are grouped into two categories. One strategy of one category uses the distance between simulated artificial doublets and the observation cells to identify doublets. For example, DoubletFinder[5] adopts this strategy to handle the doublet detection task as a binary classification problem. The other strategy used by cxds in the scds[15] package is based on co-expressed 'marker' genes that are not simultaneously expressed in the same singlet cell but can appear in doublet cells. However, the performance of existing computational approaches for doublet detection varies greatly in overall detection accuracy, impacts on downstream analyses and computational efficiency[7]. Here, we describe a strategy based on an ensemble algorithm of machine learning for doublet identification. Our approach, Chord, integrates three representative computational doublet detection methods in R environment (Supplementary Table 1), including DoubletFinder[5], bcds and cxds[15], to enhance the improvement in doublet detection (Fig. 1a).

The Chord workflow is composed of three main steps (Fig. 1a). (i) Generating training data after coarse removal of doublets using existing methods and generating artificial doublets from the filtered data. (ii) Generalized Boosted Regression Modeling (GBM) model[16] fitting, which integrates and weights the predictions of existing doublet detection tools based on classification performance on the training data. (iii) Application of the trained GMB model to the original dataset to predict doublets.

Doublets in the original dataset might cause two types of potential errors which may be introduced into the training set: (i) In the process of generating doublets, the doublets will also be treated as singlets to simulate new doublets, resulting in wrong doublets introduced into the training set. (ii) In the generated training set, the remaining doublets in the original data will be marked as singlets. Therefore, the Chord first roughly estimates the doublets of the input droplet data according to the three built-in methods: DoubletFinder, bcds, and cxds to filter out the likely doublets from the original data before simulating artificial doublets. We called this step "overkill". Selecting "overkill" could improve the accuracy of training sets, which is beneficial to model fitting (Methods; Supplementary Fig. 1c). Next, a simulation training set is generated from quality singlet data after removing these likely doublets. After evaluating the simulation training set using DoubletFinder, bcds, and cxds to get their predicted scores, the GBM algorithm was adopted to integrate these predicted scores, which served as the predictors in the GBM model. Then the doublet scores output was calculated by the GBM model for the input droplets data (Supplementary Fig. 1a).

To determine whether the ensemble algorithm improves the performance of doublet detection, we first evaluated these methods on ground-truth scRNA-seq datasets that label doublets using the experimental strategies demuxlet[11] and Cell Hashing[9]. The regions of ground-truth doublets in UMAP show enrichment of the Chord's doublet scores (Fig. 1b, c). The performance results of Chord in each dataset and the average across datasets were evaluated using receiver operating characteristic (ROC) curve analysis and precision-recall(PR) curve analysis (Table 1 and Supplementary Data 1). Chord achieved the highest areas under the ROC curves (AUCs) and the highest area under the PR curve (AUPRC) value on HTO8 dataset (0.815 and 0.599) and DM-A dataset (0.831 and 0.394) respectively (Supplementary Fig. 1b, Supplementary Table 3, Supplementary Table 4, Supplementary Table 5). When the real doublet rate was taken as cutoff, Chord detected 1596 doublets in the HTO8 dataset and 56 doublets in the DM-A dataset, which were higher than any individual built-in method (Supplementary Fig. 2c).

Furthermore, to thoroughly evaluate the performance of Chord, Chord without overkill step, and the individual built-in methods under a wide range of doublet rates, we used random sampled singlets and doublets in the dataset to build a doublet

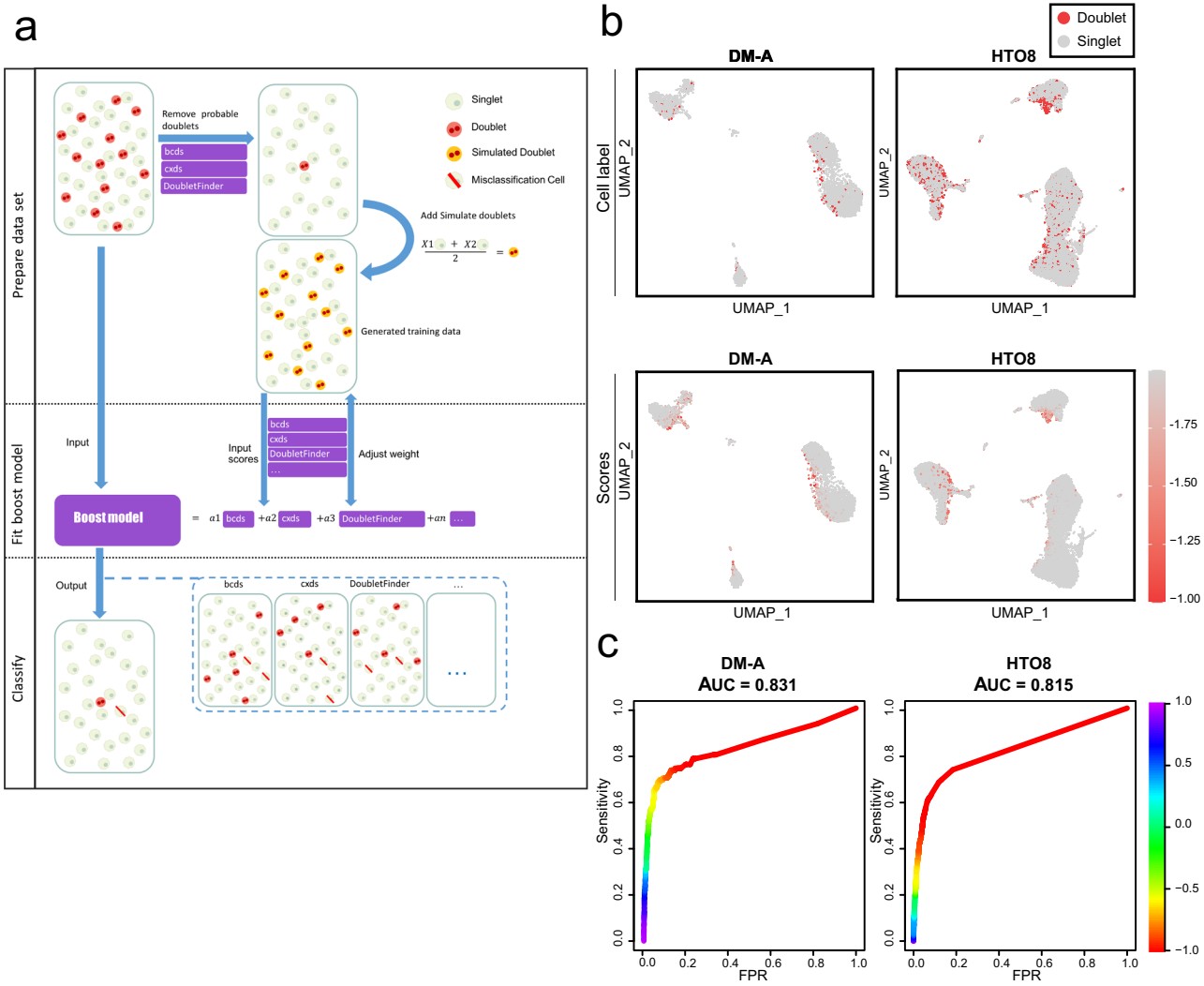

**Fig. 1 Chord overview and its performance on the DM-A and HTO8 tests. a** Schematic outline of the Chord workflow. First, preliminarily predicted doublets are filtered using bcds, cxds and DoubletFinder, and then the processed dataset is randomly sampled to generate simulation doublets that are added to the training dataset. The second step is to fit the weights of the integrated methods through the GBM algorithm on the training dataset. In the third step, the ensemble model is used to evaluate the original expression matrix and the doublets are identified by the expectation threshold value. **b** UMAP was embedded for the DM-A and HTO8 tests with experimental doublet labels. The doublets are shown in red, and the singlets are shown in grey. The doublet prediction scores of Chord were visualised on the UMAP plots for the DM-A and HTO8 tests. The DM-A dataset was from human peripheral blood mononuclear cell (PBMC) samples using the experimental demuxlet method to annotate doublets[11]. The HTO8 dataset was from the samples of PBMC using eight barcoded antibodies to mark and label doublets[9]. **c** The ROC curves of Chord were drawn for the DM-A and HTO8 tests using the R package PRROC[29].

| | PAUC800 | PAUC900 | PAUC950 | PAUC975 | AUC | PR |
|---|---|---|---|---|---|---|
| bcds | 0.598456581 | 0.697609168 | 0.747471204 | 0.772440146 | 0.797428571 | 0.465471429 |
| Chord | 0.602189241 | 0.701382997 | 0.751248502 | 0.77621856 | 0.801214286 | 0.464642857 |
| ChordP | 0.614164051 | 0.713609623 | 0.763485434 | 0.788453732 | 0.8132 | 0.466514286 |
| cxds | 0.576279854 | 0.675031498 | 0.724828357 | 0.749788808 | 0.774785714 | 0.367342857 |
| doubletCells | 0.396983835 | 0.487445535 | 0.535271112 | 0.559821949 | 0.584685714 | 0.173985714 |
| DoubletDetection | 0.569370526 | 0.666356657 | 0.715610186 | 0.740426505 | 0.793114286 | 0.500857143 |
| DoubletFinder | 0.537830717 | 0.636466839 | 0.686221775 | 0.711172069 | 0.736171429 | 0.339428571 |
| Scrublet | 0.564203075 | 0.663581473 | 0.713449225 | 0.738419459 | 0.763414286 | 0.399771429 |
| Solo | 0.604236942 | 0.703224153 | 0.752987364 | 0.777930394 | 0.803142857 | 0.434714286 |

**Table 1 Comprehensive performance of each method in real-world scRNA-seq datasets with experimentally annotated doublets.**

The average performance of various methods in all datasets. The indexes are the pAUC800, pAUC900, pAUC950, pAUC975, AUC and AUPRC.

rate gradient (Methods). The performance of these methods generally showed an upward trend as the doublet rate increased, except that the DoubletFinder had dropped significantly at some doublet rates. In the doublet rate gradient test of the random sampled DM-A dataset, bcds had the best performance while the effect of Chord was only inferior to that of bcds. In the 15 datasets with doublet rates ranging from 2 to 30% generated from the random sampled HTO8 dataset, Chord ranked first in terms of the AUC and AUPRC most times (Supplementary Fig. 1c). Chord outperformed other methods on many doublet rates and overkill is beneficial to Chord performance.

**The performance of doublet detection approaches on ground-truth datasets.** In addition to the abovementioned computational doublet detection approaches based on the R environment, some cutting-edge doublet detection software programs based on the Python platform have also been published in recent years. To integrate more doublet identification algorithms to improve the accuracy without losing the usability of Chord and the convenience of the R environment, Chord developed an expandable port allowing integration of more doublet identification algorithms (Supplementary Fig. 1a). This port can take the scoring results of other doublet detection software as input files, integrate these new methods with the individual built-in methods and obtain a training model to further improve the accuracy of doublet identification. Combinations of doublet detection methods were evaluated and the optimal combination (Chord, Scrublet[4,6] and DoubletDetection[17]) was decided based on the mean AUC (Supplementary Fig. 3b, Supplementary Table 7). We used the Chord port to integrate the two Python software (Scrublet and DoubletDetection) (Supplementary Table 1) for an enhanced GBM algorithm model, called Chord Plus version (ChordP) (Supplementary Fig. 3b).

To compare the doublet detection performances of Chord, ChordP and the other seven stand-alone software programs, we chose the seven ground-truth scRNA-seq datasets (Supplementary Table 2) to evaluate their overall performance. Chord and ChordP achieved improved accuracy, and what's more important was that it showed stability across datasets (Fig. 2). Compared with Chord, the AUC of ChordP increased from 0.831 to 0.833 on the DM-A dataset and from 0.815 to 0.835 on the HTO8 dataset (Fig. 2a), and ChordP performed better on most datasets. Moreover, partial areas under the ROC curve (pAUC) at 80% (pAUC800), 90%(pAUC900), 95%(pAUC950) and 97.5% (pAUC975) specificity were calculated, the average AUC, pAUC800, pAUC900, pAUC950 and pAUC975 of ChordP across all the datasets were the highest among all methods (Fig. 2d, Table 1), and its average rank value in all datasets reached the highest (2.285, Fig. 2c, Supplementary Data 2). These results showed that ChordP can indeed obtain more accurate results after ensembling 5 methods. In addition, the ranking variance of ChordP was 1.254, which was lower than that of Solo[6] (3.047) and bcds (1.773), both of which had the same high accuracy rate (Fig. 2c, Supplementary Data 2). This finding shows that ChordP has better versatility for different datasets than the other methods.

Through the uniform manifold approximation and projection (UMAP) method for visualising the true positive doublets (TP), true negative doublets (TN), false negative doublets (FN) and false positive doublets (FP) (Supplementary Fig. 2), the distributions of the doublets detected by each method were various at cluster level, which intuitively showed the complementarity between the different methods and the necessity of ensembling these methods. Among them, some methods, such as doubletDetection and DoubletFinder, had a concentrated distribution of FP results in the HTO8 dataset. The removal of doublets based on

these scoring results may lead to the accidental deletion of such FP cell-enriched clusters, affecting cell type proportion statistics and directly leading to the loss of rare cell subpopulations. In contrast, the FP results of Chord and ChordP were relatively evenly distributed, avoiding becoming independent clusters and affecting subsequent analysis (Supplementary Fig. 2). Above all, the results showed that ChordP, which integrates more algorithms, outperforms Chord and the other methods.

We tested the time consumption of these different software programs under uniform hardware conditions and found that Chord did not significantly increase the time consumption. Cxds was extremely time-efficient, while Solo was the most time-consuming method in a CPU environment (Fig. 2e, Supplementary Data 3).

**The performance of doublet detection approaches in DEGs and pseudotime analysis.** To evaluate the effect of different methods on downstream analysis, we used synthetic scRNA-seq datasets from a recent benchmarking research[7] to compare Chord and other approaches in terms of differentially expressed gene (DEG) detection and pseudotime analysis. In the DEG analysis, one of the synthetic scRNA-seq datasets was 'clean data' with two cell types and 1126 between-cell-type DEGs, while the other dataset was the 'contaminated data' mixed with doublets at a 40% doublet rate (Fig. 3d). We applied Chord and other approaches to remove the predicted doublets from the contaminated data, and each method generated a 'filtered dataset' (Fig. 3e). In the clean data, the contaminated data, and the filtered datasets obtained from each doublet detection approach, DEGs were analysed using the Wilcoxon rank-sum test[18] and model-based analysis of single-cell transcriptomics (MAST)[19]. Three accuracy measures, namely, the true positive rate (TPR), true negative rate (TNR) and accuracy, were used to evaluate the results of DEG analysis. In Fig. 3a, even on the contaminated data, all the data processed by each doublet detection approach showed extremely high TNRs on the two differential gene detection algorithms, because the two algorithms try to detect more correct differential genes instead of detecting as many differential genes as possible. The results of TNR and accuracy showed very tiny differences in different data, it might be due to the negative results as the majority in the dataset[7]. The accuracy of DEG analysis has been improved on these filtered datasets with doublet detection methods. The low TPR for the contaminated data indicated that it was more difficult to identify DEGs in contaminated data. The results of these methods were better than contaminated data, and the results of Chord, DoubletFinder and DoubletDetection are closer to the clean data results. In the pseudotime analysis, the 'clean data' is a synthetic scRNA-seq dataset including a bifurcating trajectory, while the 'contaminated data' is composed of 'clean data' plus 20% doublets (Fig. 3d). The 'filtered dataset' was generated from the contaminated data with removal of the predicted doublets by Chord and other approaches (Fig. 3e). Then, two pseudotime analysis methods (Slingshot[20] and monocle[21]) were implemented on the clean data, the contaminated data, and the filtered datasets. In the cell trajectory inferred by monocle, the contaminated data and the filtered datasets from bcds and cxds generated additional bifurcation trajectories due to the influence of doublets (Fig. 3b). The unrecognised real doublets by doubletCells[22] had a clear tendency to deviate from the trajectory distribution. In the results of Slingshot (Fig. 3c), the trajectories of the contaminated data and the filtered datasets of bcds and doubletCells obviously had one more branches. In contrast, Chord and Scrublet had similar cell trajectories to the clean data in the two pseudotime analysis methods, and there were fewer remaining doublets and no new branches were generated. Thus, we can conclude that Chord was

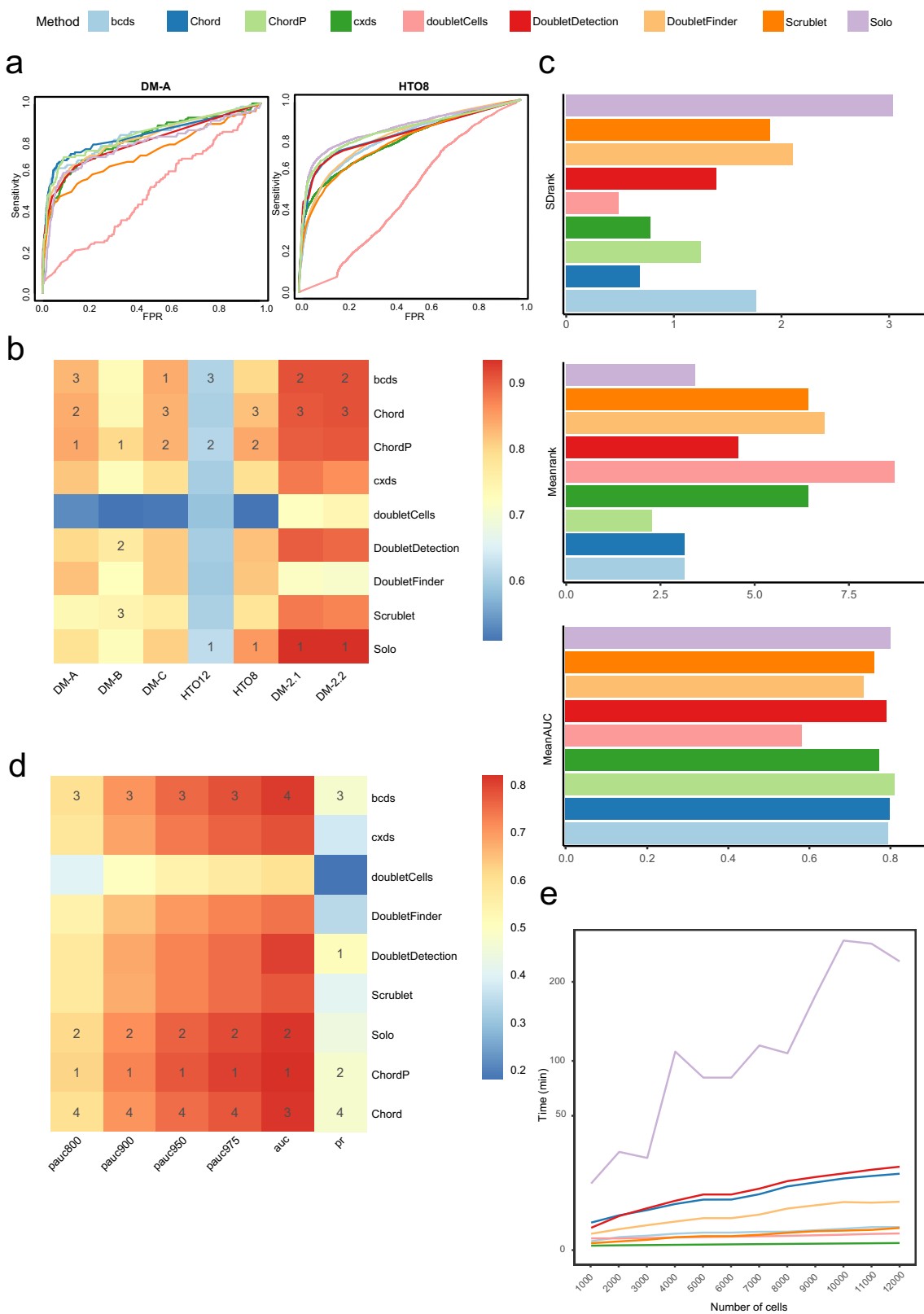

equivalent to or even outperformed other methods in DEG detection and pseudotime analysis on the synthetic scRNA-seq datasets.

**Applying Chord to real-world scRNA-Seq data.** To investigate the application of Chord in real-world data and whether the effect

of downstream analysis has been improved, we tested the Chord method on a real-world scRNA-seq data dataset without doublets labelling information that containing 52,698 cells from lung cancer tumour tissues of 5 patients[23]. Based on the expected doublet rate (0.9% per 1000 cells), we estimated the proportion of doublets in 5 malignant tumour samples (sample 11, 13, 17, 18, 22) with the most expected doublet rate (3.81%, 4.68%, 4.74%,

**Fig. 2 Comparison of the doublet detection approaches on the ground-truth datasets. a** DM-A and HTO8 tests were evaluated by nine methods and their ROC curves. **b** The AUCs of the nine methods on seven datasets and the heatmap of the AUC results. The number in the heatmap indicates the rank of the method in the dataset (only the top three methods are marked). **c** The standard deviation of the rank values for each method, the mean of the rank values of each method, and the mean AUC of each method across seven datasets. **d** Heatmap of average pAUC800 pAUC900, pAUC950 pAUC975, AUC and PR in DM-A, DM-B, DM-C, HTO12, HTO8, DM-2.1 and DM-2.2 dataset. The number in the heatmap indicates the performance rank of methods in the dataset (only the top four methods are marked). **e** By random sampling from the real DM-2.1 dataset, simulated datasets with varying numbers of cell number (from 1000 to 12,000 cells with an interval of 1000) were constructed, and the runtime of each method on the simulated datasets was recorded using the same computer server (E5-2678v3 CPU processor and 256 GB memory).

4.20%, and 4.42%), as well as the number and proportion of doublets for different cell types (Supplementary Data 4). After detecting doublets by Chord on the labelled cell from the original paper (https://gbiomed.kuleuven.be/scRNAseq-NSCLC), we can see that T cells contained the largest number of doublets due to their large cell number. The identified doublets of fibroblasts cells accounted for only 1.69%, while the proportion of myeloid was the highest (8.15%) (Fig. 4a). Doublets were unevenly distributed in the UMAP plots (Fig. 4b), clustered at the edge of some clusters, and some even formed independent clusters. According to the number of predicted doublets in different clusters, cluster 10 had the highest doublets enrichment trend (Fig. 4b, c), in which markers of T cells and plasma cells are simultaneously expressed (Fig. 4d). Cluster 10 is shown to be the closest neighbour to both cluster 1 and cluster 8 on the UMAP plot (Fig. 4b). Cluster 1 is T cell cluster, while cluster 8 is plasma cell cluster which is a cell subtype of B cells. Obviously, the doublet removal by Chord can have a great impact on the proportion of cells to avoid these imbalanced distributions, so that numerous doublets wont result in becoming noise contamination for the quantitative statistics of the proportion of cell types.

To test whether Chord is able to improve the effectiveness of downstream analysis, we evaluate the performances of Chord on real data. We utilised ROGUE[24], an entropy-based metric, to assess the purity of cell types in the original and filtered data. The ROGUE index has been scaled in the range of zero to one where the larger value means the higher purity. As a result, the ROGUE value of the filtered data was improved, the increasing trend of ROGUE value after filtering doublets was apparent (Fig. 4e). Next, we used SciBet[25], a cell type annotation tool based on Bayes decision, to annotate the cell types from the original data and filtered data, and calculated the changes in the cell types before and after applying the doublet filter. The accuracy of the annotation results of the B cells (0.726), endothelial cells (0.935), and epithelial cells (0.91) on the filtered dataset were all greater than those of the B cells (0.705), endothelial cells (0.908), and epithelial cells (0.904) in the unfiltered dataset (Supplementary Figure 3d). In addition, more differentially expressed genes can be found after Chord processing, indicating doublet removal can improve the effect of DEG analysis (Fig. 4f). Since a doublet was caused by multiple cells with the same barcode, doublet cells generally contain a higher unique molecular identifier (UMI). The UMI of doublets detected by Chord was significantly higher than singlets in all cell types (Fig. 4g). Since myeloid has the highest proportion of doublets (8.15%) and have a biological rationale in tumour microenvironment[26]. Myeloid cells were selected to demonstrate that Chord is able to correct the direction of the cell trajectory. In the pseudotime of myeloid cells in the dataset, the doublets were unevenly distributed in the dimensionality reduction plot and aggregated on the right side. After filtering the doublets, the direction of the cell trajectory changed, which might be closer to the real situation (Fig. 4h). We inferred that the deviation was corrected by removing the doublet data.

Therefore, we believe that Chord's doublet processing of real data can improve the purity of cell clusters, allowing researchers to obtain more accurate cell type identification, accurately identify DEGs between cell types and obtain better pseudotime analysis results.

## Discussion

A number of tools have been developed to remove doublets from scRNA data, but most of them cannot perform consistently well on all datasets (Fig. 2b). For users, it is difficult to evaluate and choose the most suitable method. To solve this problem, Chord integrates the results of different methods through the GBM algorithm. The benefits of each method are retained, and the disadvantages are minimised. According to our evaluation, Chord, with a high average ranking and stability, is widely applicable to various datasets and is able to integrate the doublet prediction scores from any method. It can accept any updates to the existing approaches and it will be compatible with any new approaches in the future (Supplementary Fig. 1a). As novel methods continue to emerge, the better ones can always be selectively integrated to improve Chord's accuracy (Fig. 2b, c). It will be compatible with some new approaches in the future, and it can accept any update to the existing approaches (Supplementary Fig. 1a).

Doublets contained in scRNA data affect not only the quantity of cell types but also the accuracy of downstream analysis. The data filtered with Chord were accurately identified in cell type annotation (Supplementary Fig. 3d), and more potential DEGs (Fig. 4f) were found with Chord than with other methods. These results can help researchers obtain more accurate results and conclusions in subsequent analyses.

We also optimised the construction of simulated training sets. In this general step, most doublet detection methods add simulated doublets to real data to generate a training set. However, the training set may contain undiscovered doublets, which could limit the training of doublet detection models and reduce accuracy. Therefore, Chord includes "overkill" step. First, built-in methods are used to evaluate the data. Then, filtered doublets identified with various methods are removed, and doublets simulated based on the remaining cells are used to generate a training set. In this way, the number of undiscovered doublets in the training set is greatly reduced, thus improving the accuracy of training and doublet detection. The use of an "overkill" step might also improve the performance of other methods.

Chord requires the integration of the results of multiple methods, so it is not optimal in terms of time efficiency. The use of refactoring the integrated methods may solve this problem. Also, other pre-processing methods, e.g., ambient mRNA removal steps for droplets, should be considered to improve the accuracy of downstream analysis[27].

Overall, we proposed a computational approach for doublet detection that utilizes an ensemble algorithm model. This is the first study of its kind to use an ensemble algorithm for doublet detection. This work could help researchers concisely and efficiently remove doublets from scRNA data.

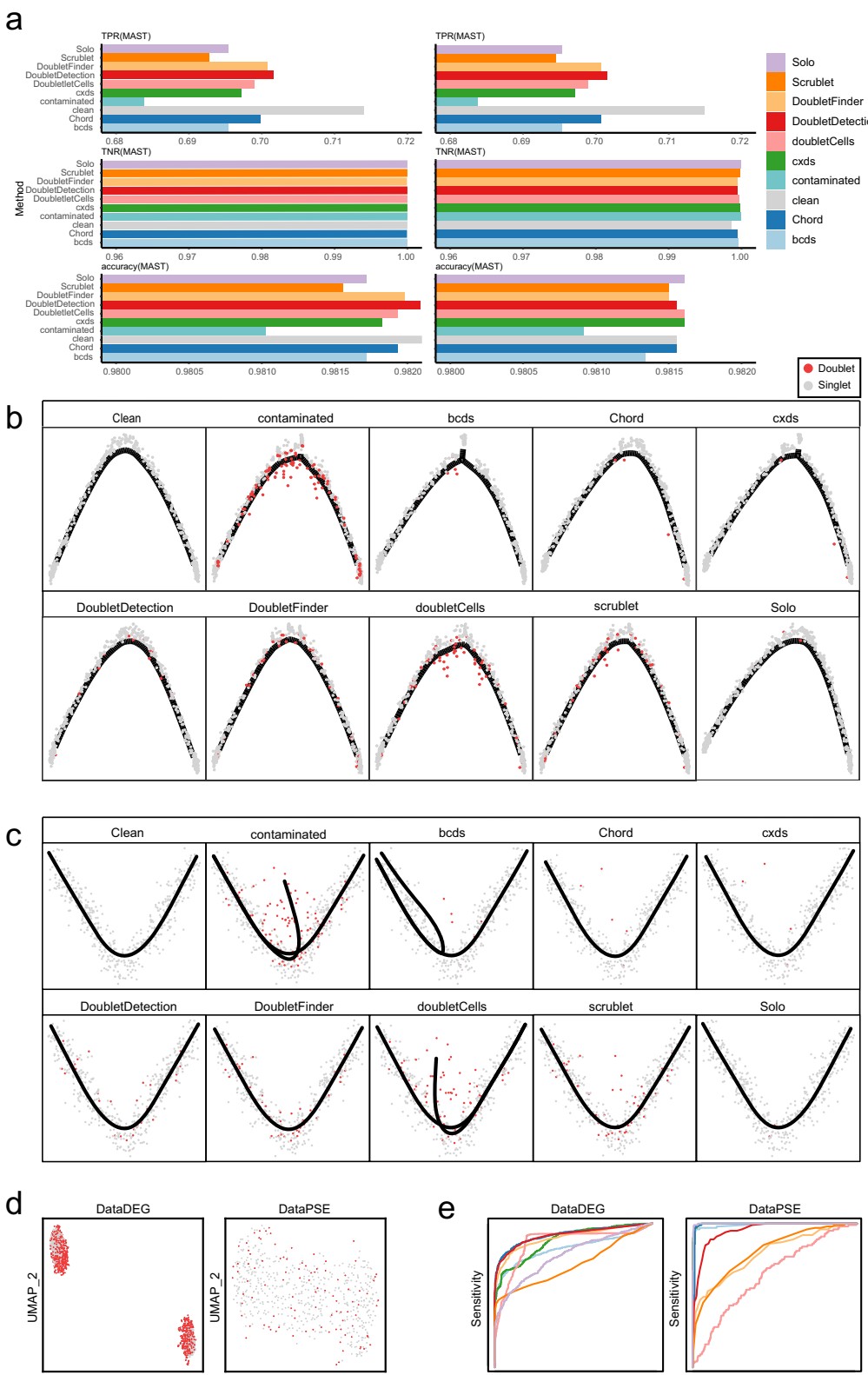

## Methods

**Chord overview**. Data input. The input format used in Chord was a comma-separated expression matrix, which was a background-filtered, UMI-based count matrix for a single sample. Chord pre-processes the count expression matrix according to the Seurat analysis pipeline. Chord can also directly accept object files generated by the Seurat analysis pipeline. In addition, it is suggested that users estimate a doublet rate (Supplementary Fig. 1) based on the loading conditions so that Chord can simulate a simulated training set that is similar to the real dataset

selected. It should be mentioned that the doubletrate, which is the estimated doublet rate parameter, has generally robust, and a certain degree of deviation will not greatly affect the results of Chord (Supplementary Fig. 3e, Supplementary Table 8).

Data pre-treatment. To train the results of bcds, cxds, DoubletFinder, and DoubletCell based on ensemble learning, Chord generates SingleCellExperiment object data conforming in the input format of bcds and cxds through the R package SingleCellExperiment[28].

**Fig. 3 Evaluation of the doublet detection methods using the realistic synthetic datasets on DEG analysis and pseudotime analysis. a** The dataset of labelled DEGs was processed by each doublet detection method, and the top 40% of cells based on the doublet score were excluded. Then, the DEGs were detected using MAST[19] and Wilcoxon rank-sum tests[18]. Taking the DEGs as positive, three accuracy measures (i.e., the TPR, TNR and accuracy) were calculated. **b**, **c** After processing the dataset for the pseudotime analysis using each doublet detection method, the top 20% of cells according to the doublet score were excluded. Monocle (B) Slingshot (C) were used to construct the trajectories of these results. **d** The UMAP was embedded for the two realistic synthetic datasets (DataDEG and DataPSE), in which the doublets are shown in red and the singlets are shown in grey. DataDEG is a simulation dataset containing two synthetic cell types, including 1667 cells, 40% of which are correctly labelled doublets. DataPSE consists of 600 cells, 20% of which are synthetic labelled doublets containing a bifurcating trajectory. **e** The AUC of each method on DataDEG and DataPSE and their ROC curves.

Preliminary deletion of doublets. We tend to reduce the number of doublets in a sample as much as possible before generating simulated doublets to avoid training datasets that contain real doublets because real doublets are not preliminarily labelled and are thus labelled as singlets in the training set. We applied three doublet detection methods, evaluated the selected datasets, and roughly filtered the doublets.

We used the scds() function, cxds, and bcds to evaluate the data based on the parameters ntop = 500, binThresh = 0, and retRes = T; then, we extracted the doublet scores obtained with the two methods for each cell. Next, we used the no ground-truth process of DoubletFinder for evaluation. The parameters were set as PCs = 1:10 and pN = 0.25, and automatically extracted the pk value corresponding to the highest bimodal coefficient, and obtained the doublet score.

Chord introduced an adjustable parameter called overkillrate. According to this paremeter, we filtered the doubletrate*overkillrate percentage of cells that were most likely to be doublets according to the evaluation results of each method and obtained the 'prefiltered data'. By default, we set this parameter to 1 to exclude the doublets identified by the three built-in methods at the selected doublet rate.

Generating the simulation dataset. To avoid the generation of doublets synthesised from the same cell type, Chord randomly sampled pairs of cells in the pre-filtered data, generated simulated doublets from the raw UMI count by mixing the gene expression profiles of the selected cell pair[4] and then added simulated doublets to the pre-filtered data:

1. Perform a Seurat standardisation process on the data and call the functions NormalizeData(), FindVariableFeatures(), and ScaleData() with default parameters.
2. Cluster cells, perform dimensionality reduction operations on the data with RunPCA(), taking PC1 to PC30 as inputs, and perform k-means clustering (k = 20). After clustering, the cells were divided into 20 clusters.
3. Randomly sample pairs of cells at a ratio of doubletrate/(1-doubletrate) for each cell type, and weight the cells with introduced biological random number from a N(1, 0.1) distribution which was set to roughly represent experimental randomness.
4. Average the weighted gene expression profiles of the cell pair as simulated doublets.
5. Add the simulated doublets to the pre-filtered data; take this new dataset as the training set.

Model training. For the training set,in which all cells were labelled, Chord used the same parameter settings to evaluate the doublet scores through the bcds, cxds, DoubletFinder methods. GBM (from R package gbm) which performed better than AdaBoost, XGBoost, and LightGBM (Supplementary Fig. 3a) was used to combine the prediction scores of the built-in methods to fit a model for robust estimation. In GBM, each individual model consists of classification or regression trees, also called boosted regression trees (BRTs). We defined 1000 trees for fitting, and set parameter shrinkage = 0.01and cv.folds = 5. The function DBboostTrain() was defined to implement model training, and it combined the scoring results of these built-in methods into a matrix. Then the matrix was input data into the function gbm() in the R package gbm. The simulated doublets were set as true positives (TPs), and the singlets were set as true negatives (TNs) for model training.

Scoring the original data. We defined DBboostPre(), used the model trained on the training set to predict the doublet scores of the original dataset, and output the doublet score of each cell based on the result of the ensemble model.

Expandable interface. To incorporate version updates of the integrated doublet tools and the release of new doublet tools, Chord included an extendible interface that could be customised based on the doublet evaluation results of any tool. To incorporate the selected methods, at first the chord() function was used to extracted the expression matrix after generating the simulated doublets. Next, we exported prefiltered data, evaluated both the original and prefiltered data using the selected methods, and then imported the scores into Chord. Based on the evaluation scores, Chord used the GBM algorithm to the extra methods together with bcds, cxds and DoubletFinder. At last the ensemble model was used to score the original dataset.

**DoubletCells settings**. DoubletCells from the R package scran 1.16.0 with the parameters k = 50 and d = 50 was selected and ran in the R 4.0.2 environment.

**Solo settings**. Solo was run in the Python 3.7.9 environment, and the parameters were set according to the reference file solo_params_example.json downloaded from https://github.com/calico/solo. The doublet scores of each cell were read through the softmax_scores.npy file.

**DoubletDetection settings**. Double detection was run in the Python 3.7.9 environment using the operating parameter settings from https://nbviewer.jupyter.org/github/JonathanShor/DoubletDetection/blob/master/tests/notebooks/PBMC_8k_vignette.ipynb. Scoring results were obtained with BoostClassifier.fit.voting_average_.

**Scrublet settings**. Scrublet was run in the Python 3.7.9 environment, based on the instructions at https://github.com/AllonKleinLab/scrublet/blob/master/examples/scrublet_basics.ipynb. The doublet scores of each cell were read through Scrublet.scrub_doublets.

**Doublet rate gradient**. The doublet gradient dataset was composed of random samples from real datasets. The HTO8 dataset was randomly sampled from the DM-A dataset with an interval of 0.02, and the ratio of the doublet rate was baried from 0.02 to 0.30. DM-A dataset was randomly sampled from the DM-A dataset at an interval of 0.01, and the ratio of the doublet rates was baried from 0.01 to 0.10.

**Rank variance comparison for different methods**. Because of the unstable performance of the methods for different datasets, we calculated the AUC rank variance coefficient to characterise the stability of each method based on different datasets. The methods used in the evaluation, were ranked according to their AUC results, and the variance of the AUC ranking of each method was calculated for different datasets as the AUC rank variance coefficient (SDrank). The method with the highest SDrank was unstable for different datasets, it represented the generalisability of methods.

**AUROC and AUCPR calculations**. Each method was evaluated using the AUC with ground-truth labels for the original datasets or labels for simulated datasets. In addition, the AUPRC and partial area under the ROC curve (pAUC) were calculated. We calculated the AUC and AUPRC with the PRROC R package[29], and plotted the ROC curves for individual method by setting the option 'ROC curve' to 'TRUE'. For the pAUC at 0.9, a 0.95 specificity value was calculated by using the 'pROC'[30] R package.

**Evaluating DEGs with simulated datasets**. We used the published simulated single-cell sequencing dataset[7] to test the changes in the number of DEGs correctly detected before and after doublet removal by different methods. The simulated single-cell dataset was generated with scDesign[31] and contained 1,667 cells and 18,760 genes. It was divided into two cell types each counted 500 cells, and 667 doublets simulated by those singlets. Among them, high-expression and low-expression DEGs, which were known at the time of data generation, accounted for 6% of all sample (3% upregulated genes and 3% downregulated genes). The dataset without doublets was used as the clean dataset, and the data with doublets were added as the contaminated dataset. After the contaminated dataset was evaluated with a doublet detection method, the dataset of 40% of the cells with the highest score was filtered according to the result. We performed the process described above for each method. Then, we used Seurat's FindAllmarkers() function with the methods 'wilcox'[18] and 'MAST'[19] to perform DEG calculations on the dataset. In order to find DEGs, we removed genes with fold changes below 0.25; then, genes with Bonferroni-corrected p values below 0.05 and genes detected in a minimum fraction of 10% cells in either of the two clusters were defined as DEGs. Finally, we calculate the accuracy, TPR, and TNR for all datasets.

**Pseudotime analysis of the simulated data**. We used the published dataset of simulated single-cell sequencing[7] to test the effects of different methods on pseudotime analysis. The simulated single-cell dataset, which was generated with Splatter, consisted of 600 cells and 1000 genes. There were two cell tracks containing 250 simulated cells and 100 simulated doublets. The dataset without the doublets was used as a clean dataset, and the data containing the doublets were used as contaminated dataset. After evaluating the contaminated dataset through a doublet detection method, cells predicted to be doublets were deleted. After then,

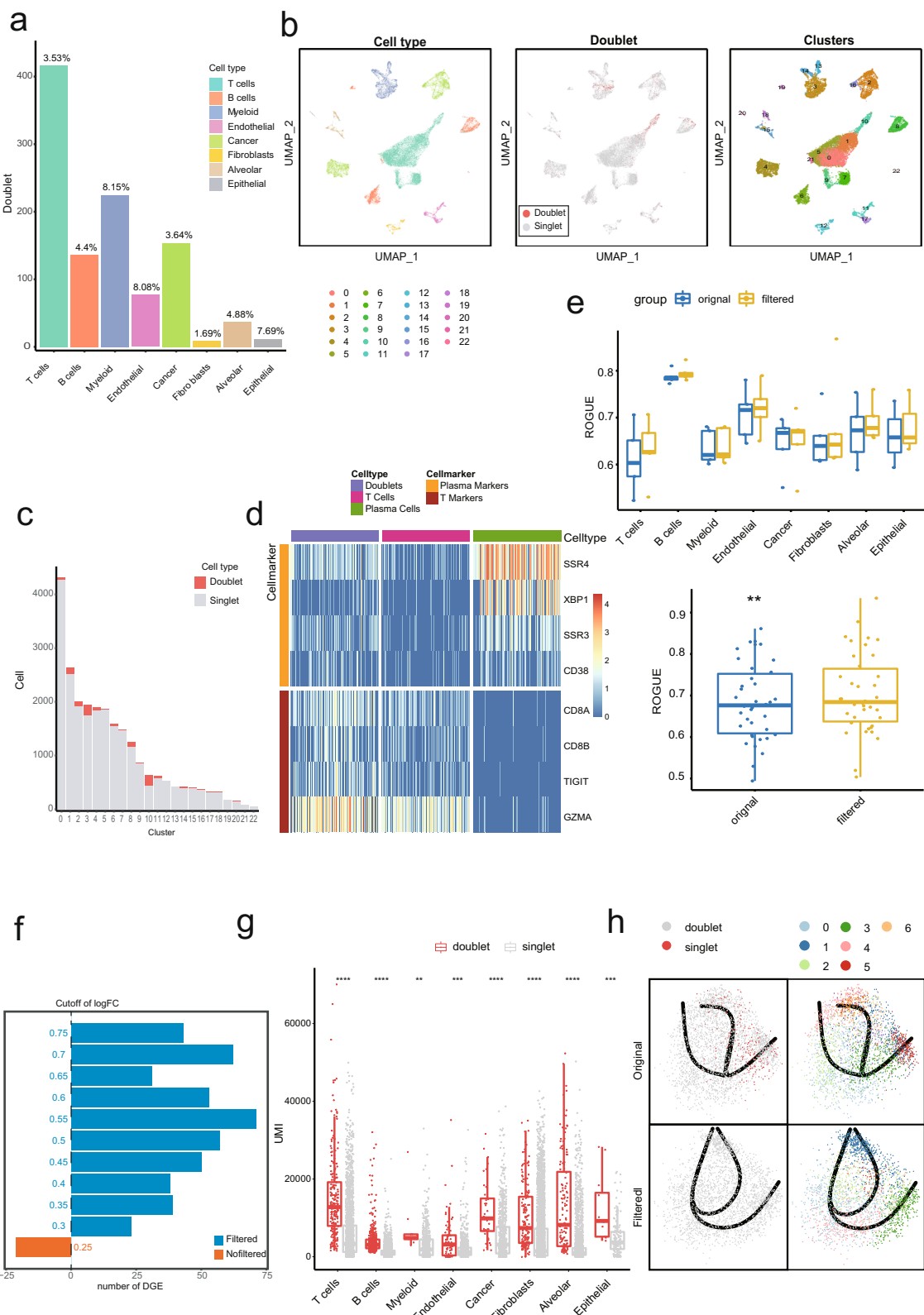

the R package monocle and Slingshot software were used for a pseudotime analysis of the dataset. We implemented the process described above for each method. In addition, we calculated the trajectories of the clean and contaminated datasets.

**Time cost**. Based on random sampling from the real DM-2.1 dataset, we constructed test datasets of 1,000 to 12,000 cells with a gradient of 1,000 cells and tested various methods with the same processor. Then, the runtime of each method

for the simulated datasets on the same computer server (E5-2678v3 CPU processor and 256 GB memory) was recorded.

**Base analysis process for lung cancer data**. We obtained the Seurat object from SCope (https://gbiomed.kuleuven.be/scRNAseq-NSCLC) including 19 samples from five patients.

**Fig. 4 Doublet removal by Chord improves the analysis performance on real-world scRNA-Seq data. a** Doublet detection was performed on the published lung cancer dataset[23] using Chord. The number and the proportion of doublets for each cell type which was labelled by original paper were recorded. **b** UMAP of the 24,280 cells in this dataset. The cells were coloured by cell type (left), the predicted result of doublet detection (middle) and the clusters were defined by Seurat (right). **c** A bar chart showing the number of doublets and total cells in each cluster. **d** Heatmap of marker genes for doublets (in cluster 10), T Cells, and Plasma cells. **e** The ROGUE value[24] of each cell type for each sample. A paired t-test was used to test the difference in each cell type in each sample between the two groups before and after doublet filtration (paired t-test, $*p < 0.05$, $**p < 0.01$, $***p < 0.001$, $****p < 0.0001$). **f** A bar chart showing the changes in the total number of differentially expressed genes before and after doublet removal. The DEGs were calculated by the Wilcoxon rank-sum test (Seurat). The threshold value of logFC was measured by a gradient from 0.25 to 0.75 at 0.05 intervals. **g** The RNA UMI numbers predicted by Chord in each cell type were significantly different between the doublets and singlets (unpaired t-test, $*p < 0.05$, $**p < 0.01$, $***p < 0.001$, $****p < 0.0001$). **h** The myeloid cells in original and filtered data for the pseudotime analysis were processed using Slingshot[20]. The trajectory of the original data and the filtered data are shown respectively.

After evaluating the predicted double cell rate for each sample based on the number of cells, we selected the 5 tumour samples (sample 11, 13, 17, 18, 22) with the highest predicted double cell rate and applied Chord individually to each sample. Then we performed a standard Seurat analysis:

1. The expression matrix and metadata for samples 11, 13, 17, 18 and 22 (33694 genes across 24280 cells) was extracted, and a new Seurat object was created.
2. After the data were normalised, 2000 variable genes were screened with the function FindVariableFeatures().
3. PCA was used to reduces the dimension of the data to 50 dimensions, and PC1 to PC30 were used for clustering. Through the FindNeighbors function (resolution = 1.5), we divided the cells into 22 clusters, and we computed the UMAP embeddings to display the results[23].

**Cluster purity**. We calculated the rogue() (R package ROGUE) value for each cell type in each sample. Comparisons between two original datasets and filtered datasets were performed using paired two-tailed t-tests.The parameters of ROGUE were set as "platform = UMI" and "span = 0.6".

**Cell type identification using SciBet**. We used the function SciBet() (R package SciBet[25]) to perform cell type analysis on epithelial, endothelial, myeloid, T, and B cells before and after processing. The reference of human cell types were provided by SciBet (http://scibet.cancer-pku.cn/major_human_cell_types.csv).

**Calculated number of DEGs**. The number of DEGs were calculated by the Wilcoxon rank-sum test (Seurat) with the following parameters: min.pct = 0.1, and test.use = 'wilcox', additionally, and the threshold of logFC was varied from 0.25 to 0.75 at an interval of 0.05. Then, we counted the number of DEGs at different logfc.threshold values.

**Statistics and reproducibility**. The details about the steps and statistics of each analysis were recorded in each part of methods. Statistical analyses were performed using R package ggpubr (https://rpkgs.datanovia.com/ggpubr/) or build-in function of R 4.0.2 (https://www.r-project.org/). A P value less than 0.05 was considered as statistically significant.

**Reporting summary**. Further information on research design is available in the Nature Research Reporting Summary linked to this article.

## Data availability
The data analysed during this study were collected from public studies or databases. The access to data was shown in Supplementary Table 9.

## Code availability
The Chord software package, including documentation, tutorials is available at https://github.com/13308204545/Chord[32].

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

## Acknowledgements
We thank Dr. Yong Bai from BGI-Shenzhen for the algorithm suggestions. This research was supported by the Guangdong Enterprise Key Laboratory of Human Disease Genomics (2020B1212070028), and Shenzhen Key Laboratory of Single-Cell Omics (ZDSYS20190902093613831).

## Author contributions
H.L.Z. and K.X.X. designed the research, performed the data analyses, and wrote the codes and the manuscript. K.K., J.H.Y., and H.M.Y. made suggestions to optimise the article. C.L. revised the manuscript and provided suggestions to data analysis. G.B.L. provided critical advice and oversight for the research and revised the manuscript. All authors have read and approved the final manuscript.

## Competing interests
The authors declare no competing interests.
