## [Peer Review File · Communications Biology]

Reviewers' comments:

Reviewer #1 (Remarks to the Author):

The authors describe Chord – a computational pipeline for scRNA-seq doublet detection that leverages an ensemble approach to aggregate the results of existing doublet detection tools. The authors begin by providing an overview of the Chord workflow, which includes three key steps: (i) Generation of training data after coarse removal of doublets using existing methods and generation of artificial doublets from the “cleaned” data, (ii) Adaboost model fitting which integrates and weights the predictions of existing doublet detection tools based on classification performance on the training data, and (iii) Application of the trained adaboost model to the original dataset to predict real doublets. We find the authors description of the Chord workflow to be well-considered and a notable contribution to the single-cell genomics doublet detection literature. However, we have concerns regarding the choice of doublet detection methods used during training data generation (Major Comment #1) and adaboost model training (Major Comment #2), as well as concerns regarding documentation of scRNA-seq pre-processing and doublet method parameter fitting (Major Comment #3).

After describing the Chord workflow, the authors benchmark Chord against the four ‘built-in’ doublet detection methods used to generate the training data (e.g., DoubletFinder, doubletCells, bcde, and cxd) on two publicly-available PBMC scRNA-seq datasets where doublets are empirically-defined using cell hashing or in silico genotyping. This analysis reveals that Chord performs with similar accuracy and stability to existing doublet tools on the full datasets, as well across datasets where doublets are proportionally up- and down-sampled. The authors then repeat this benchmarking workflow for Chord-plus (ChordP), which incorporates additional doublet detection methods (e.g., DoubletDetection, Solo, and Scrublet) into the Chord adaboost training step. This analysis reveals that including more doublet detection methods into the adaboost training step improves ChordP performance relative to Chord, although improvements over existing doublet detection methods remain modest. Altogether, we found these benchmarking workflows to be well-executed.

Having established that Chord and ChordP are amenable to scRNA-seq doublet detection, the authors then use simulated scRNA-seq datasets from Xi & Li (Cell Systems, 2020) to benchmark their method during pseudotemporal ordering and differential gene expression analyses (computational tasks known to be improved after doublet removal). This work reveals that Chord performs comparably to other doublet detection tools, although we disagree strongly with some of the claims the authors make in this section of the manuscript (Major Comment #4).

Finally, the authors apply Chord to ‘real-world’ scRNA-seq data of primary lung tissue from Lambrechts et al (Nature Medicine, 2018) where doublets are unknown, and demonstrate how Chord improves scRNA-seq analysis workflows downstream of doublet removal such as cell type purity (ROGUE), cell type identification (SciBet), differential gene expression analysis, and pseudotemporal ordering. While we appreciate the purpose of this analysis, we have serious concerns about how the authors applied Chord to these data (Major Comment #5) and believe that the authors should repeat this workflow on a scRNA-seq dataset that is more amenable to their purposes.

On balance, we consider the theoretical underpinnings of Chord to be sound and sufficiently novel, and find that the authors’ benchmarking workflows were well-executed. Moreover, while Chord does not perform better than existing doublet detection methods with regards to absolute accuracy or improvements in scRNA-seq analyses downstream of doublet removal, we are convinced by the authors’ claims that Chord will perform more robustly across diverse scRNA-seq datasets due to its aggregated nature. For these reasons, we believe this manuscript should be conditionally-accepted for publication in Communications Biology, assuming that our Major Comments are adequately addressed.

Major Comments

1. In both Chord and ChordP, the authors use “representative” computational doublet detection methods to remove doublets during the training data generation step. Specifically, they use the kNN classifiers DoubletFinder and doubletCells, as well as the marker gene co-expression classifiers bcds and cxds. While it is true that these are two distinct types of doublet detection tools, there are other doublet detection tools that could also be incorporated to make this step truly “representative”. For example, Solo and DoubletDetection use neural networks and the hypergeometric test to classify doublets in a fashion that is distinct from bcds, cxds, DoubletFinder and doubletCells. Moreover, it is clear from the authors’ own analyses that doubletCells performs quite poorly across most of the benchmarking datasets, which begs the question why doubletCells is relied upon at all for training data generation.

One critical analysis that we encourage the authors to include in the revised manuscript is a systematic analysis of Chord performance (e.g., mean AUC across benchmarking datasets) using all combinations of doublet detection methods for training data generation. Is there an ideal combination of methods to use for this step? Are there methods that decrease performance when they are included?

2. In a similar vein as Major Comment #1, we are curious whether further optimization can be introduced with regards to which models are included in the adaboost training step. We see that ChordP improves upon Chord as more models are used – however, does this improvement represent the optimal performance of the method? Moreover, does inclusion of more models with more similar underlying algorithms bias the aggregated Chord result (in a fashion that reflects biases amongst the existing doublet detection literature)?

We encourage the authors to repeat the analysis suggested in Major Comment #1 but on the adaboost training step to identify whether Chord performance is sensitive to which sets of models are included.

3. The authors note in the Methods section that Chord uses default parameter settings for the aggregated doublet detection models. However, it is key to note that some doublet detection models require parameter optimization for effective performance. For example, the authors implement DoubletFinder “with parameter selection PCs = 1:10, pN = 0.25, and automatically extracting the pk value corresponding to the highest bimodal coefficient to extract the doublet score.” It must be noted that DoubletFinder is implemented to interface with a pre-processed Seurat object wherein the ideal number of PCs are determined by the user for their particular dataset before using the method. That is, hard-setting PCs = 1:10 will necessarily result in sub-optimal DoubletFinder (and, as a result, Chord) performance. Moreover, it is not uncommon for mean-variance-normalized bimodality coefficient distributions to exhibit multimodality (this often happens when data is not properly quality-controlled), which in turn requires DoubletFinder users to manually interrogate the predictions after using distinct pK values to identify the “correct” parameters. As a result, automatically setting pK to the maximum bimodality coefficient value may further cause Chord performance to deteriorate.

Thus, we think it is very important that the authors build out the pre-processing guidelines for scRNA-seq data prior to using Chord, as well as parameter optimization guidelines for each doublet detection method used in their aggregated pipeline.

4. While we found the differential gene expression analyses and pseudotemporal ordering comparisons to be well-executed, the authors posit a perplexing explanation for the increase in detected DEGs relative to “clean” synthetic data following doublet removal by some doublet detection methods. Specifically, the authors state in lines 194-197: “...TPRs of the datasets processed by some doublet detection methods were higher than those for the clean data, which may be due to the deletion of transition state cells between cell types that were identified as doublets through these methods, resulting in obvious differences between cell types to detect

more DEGs.”

This claim is confusing because the synthetic data used in Xi & Li (and this paper) does not include any “transition state cells”, as is evident in Fig. 3D of this manuscript. Please provide an explanation as to why we are incorrect in this interpretation, or remove this claim from the manuscript. Moreover, the authors should cite Xi & Li for the description of why TNR is high across all datasets.

5. The authors use the primary lung scRNA-seq data from Lambrechts et al to demonstrate the utility of applying Chord to ‘real-world’ scRNA-seq data. According to the associated Nature Medicine manuscript, this dataset was generated by running 19 distinct 10x Genomics microfluidic lanes representing normal lung tissue and tumor core/middle/edge tissue from 5 distinct patients, each with unique cellular compositions and cell loading densities. The explanation of how this data was analyzed is a bit sparse – so apologies if we are interpreting your work incorrectly – but was Chord run on the aggregated dataset including all of these samples? Or was Chord run independently on each sample prior to combining each sample? If it is the former, then Chord will be attempting to find doublets that could not possibly exist (e.g., tumor-tumor doublets from distinct patients; tumor-endothelial doublets from normal and tumor samples, etc.) which would seriously confound these results. At a minimum, the authors must clarify how exactly they did this analysis – although other scRNA-seq datasets are likely to be more amenable to the points the authors intend to make (which we are confident will hold up).

Minor Comments

1. Paper needs significant editing for clarity and grammar.

2. The authors should explain more clearly why removing doublets during training data generation is beneficial for Chord performance. Also, the authors should clarify the workflow for generating artificial doublets – specifically, we are confused how weighting expression profiles from randomly-selected cells with a “biological random number from a $N(0.1,1)$ distribution” avoids generating synthetic doublets derived from the same cell type. Wouldn’t it be easier to manually sample cell pairs from different gene expression clusters? What is the purpose of the random number generation?

3. Lines 111-112: “Ground-truth comparisons illustrate that Chord detects most doublets” — The authors use qualitative comparisons of UMAP embeddings to make this quantitative claim. Authors should either provide the statistics underlying the claim or change the statement (e.g., Chord-derived doublet scores are enriched in regions of gene expression space associated with ground-truth doublet classifications).

4. Lines 568-569: Authors say they benchmarked Chord on the publicly-available HTO8 dataset which is comprised of “samples of cell lines using twelve barcoded antibodies to mark and label doublets.” – According to the Seurat web-page and Stoeckius et al, HTO8 is scRNA-seq of PBMCs from 8 donors, while HTO12 is the dataset the authors are referring to in the Fig. 1 legend. The UMAP embedding looks like PBMCs, so the authors should change the text to reflect the correct dataset. The authors also need to change their references to HTO8 as “cell line” data throughout the manuscript and supplemental tables.

5. Fig. S1A: “taning data set” instead of “training data set”, “adboost” instead of “adaboost”

6. Lines 320-321: “...extracted the doublet scores evaluated by two methods for each cell: i) through the DBF() function in the called R package DoubletFinder” – we are not aware of a ‘DBF’ function in DoubletFinder.

Reviewer #2 (Remarks to the Author):

This manuscript proposes a new computational doublet-detection method based on the ensemble of multiple cutting-edge methods. The basic idea is 1) remove high-confidence doublets by four

independent methods; 2) after removal, simulate artificial doublets by adding two randomly selected cells, and merge these artificial doublets back to obtain a training set; 3) train an Adaboost model on the training set, in which each independent method's doublet score vector is one predictor, and the singlet/artificial-doublet label from step one is the target variable; 4) apply the trained Adaboost to the original dataset for doublet prediction. The authors performed various validations on real and synthetic datasets to check the method's prediction accuracy and impacts on downstream analysis. The proposed ensemble framework is novel in the doublet-detection field. It also resolves some issues in other related studies. On the other hand, there are major concerns in the manuscript in terms of methods implementation, validation, and writing. I hope the authors could fix them to improve the current work.

Method and technical details

1. In lines 307-309, the authors claimed that the performance of Chord is robust across different doublet rates yet without evidence. I suggest providing a sensitivity analysis to support this argument. Some other methods have similar parameters, for example, the number of artificial doublets in DoubletFinder 1 and Scrublet 2. The selection of those parameters has been clearly shown to be robust by sensitive analysis.
2. Line 326 introduces the overkillrate parameter and claims "we filtered the doubletrate*overkillrate percentage of cells ..." It seems that overkillrate is a numerical parameter that controls the conservativeness of preliminary doublet removal. I hope the authors could provide practical guidance on how to choose this parameter and how robust the model performance is under different values. Although the software GitHub page sets this parameter a default value 1, the authors did not specify which value they use in the analysis of the manuscript.
3. After reading the "Model training" part (line 345-351) initially, I did not understand how the Adaboost model was trained, until I checked the figure on the GitHub page. I suggest adding those figures into the manuscript and explicitly demonstrating that each method's doublet score serves as the predictor/variable in the Adaboost model.
4. The methods included in Chord need to be reconsidered. First, there is one method, scDbIFinder 3, not in the ensemble or extension part. This method was published last year and has been shown excellent performance and speed compared with others 4. Second, the doubletCells method has been shown to significantly underperform others in the previous benchmark study 5, Figure 2, and Figure S1. Actually, doubletCells was removed from the latest version of package scran (v 1.20.1), and the author of scran recommends using scDbIFinder to substitute doubletCells in that package. Therefore, I suggest using scDbIFinder instead of doubletCells as one of the four backbones. At least scDbIFinder should be included in the ChordP.
5. In the "Cluster purity" part (line 440), the author should clarify which package and function were used to calculate the ROGUE value. While I can understand by checking the GitHub page of ROGUE, it is not user-friendly without a clear illustration. The same issue applies to SciBet method, where the execution details were not shown.

Validation

1. The comparison of detection accuracy on real data is not convincing. In Table S3, Chord or ChordP only achieved two highest AUROCs and one highest AUPRC on seven real datasets. The proposed methods are not attractive if they did not show a strong advantage over single methods. This is a major concern also because the running time of Chord or ChordP will be the sum of all methods included plus the training time of Adaboost. One potential solution, as stated in the last section, is to substitute doubletCells with scDbIFinder or another better single method.
2. One of the main reasons for detecting doublets is to remove spurious clusters in the datasets. The importance of this goal has been demonstrated repeatedly in previous studies 1,2,5. While the authors evaluated the impacts of doublet-detection on the quality of true clusters, the capacity of removing spurious clusters has not been discussed. I suggest adding this analysis, either through purely simulated data or real data with artificial doublets.
3. In practice, users will choose a cutoff on doublet scores to call doublets. I suggest adding one validation, which is to calculate the precision, recall, and true negative rate under certain doublet rates on real data. This analysis can give a more comprehensive evaluation of proposed methods.

Writing

Some sections of this manuscript were either not written in the academic style or with confusion. Some examples include:

1. There are several "STAR Methods" in the manuscript, which seems a unique item in articles from Cell Press.
2. Line 122 "we used random sampling to proportionally sample singlets..." seems redundant expression.
3. Line 152 pAUCXXX was shown without definition. This may confuse readers who lack expertise in machine learning terminology.
4. Line 249 "A pseudotime analysis of the myeloid cells was conducted". Please consider rewriting this sentence.
5. Line 348 "The R package 'adabag', through the AdaBoost algorithm, DBboostTrain() is used to implement model training for training based on the scoring results of the four methods." This sentence is ambiguous. Please consider rewriting.
6. Line 370 "Doublet detection" should be DoubletDetection.
7. Line 421 "the highest scoring cells for the number of double cells according to the results were removed". Consider rewriting this sentence.
8. Line 456 should be followed by lines 458 and 459.

Reference

1. McGinnis, C. S., Murrow, L. M. & Gartner, Z. J. DoubletFinder: Doublet Detection in Single-Cell RNA Sequencing Data Using Artificial Nearest Neighbors. *Cell Syst* 8, 329-337.e4 (2019).
2. Wolock, S. L., Lopez, R. & Klein, A. M. Scrublet: Computational Identification of Cell Doublets in Single-Cell Transcriptomic Data. *Cell Syst* 8, 281-291.e9 (2019).
3. Germain, P.-L., Sonrel, A. & Robinson, M. D. pipeComp, a general framework for the evaluation of computational pipelines, reveals performant single cell RNA-seq preprocessing tools. *Genome Biol.* 21, 227 (2020).
4. Xi, N. M. & Li, J. J. Protocol for Benchmarking Computational Doublet-Detection Methods in Single-Cell RNA Sequencing Data Analysis. *arXiv [q-bio.GN]* (2021).
5. Xi, N. M. & Li, J. J. Benchmarking Computational Doublet-Detection Methods for Single-Cell RNA Sequencing Data. *Cell Syst* (2020) doi:10.1016/j.cels.2020.11.008.

Reviewer #3 (Remarks to the Author):

The manuscript, "Chord: Identifying Doublets in Single-Cell RNA Sequencing Data by an Ensemble Machine Learning Algorithm" by Xiong et al. has focused doublet detection in single-cell Sequencing datasets.

Doublets/Multiplets in single-cell datasets are one of the challenging problems in field of single-cell droplet sequencing and detecting and removing them from the analyses, can have significant effect on down-stream analyses, depending on removing False Positive or False Negative Doublets. As a result, tools that can detect these doublets/multiplets in single-cell RNA-Seq are important and will be beneficial in the field. The authors have developed a tool named Chord and ChordP, which detect the doublets with high accuracy. Especially, combining the power of many available tools and machine learning are the two important points that I think makes this an excellent tool in the field and this manuscript should be considered for publication.

I have some minor comments which are not clear in the manuscript and need clarification (at least for me).

Minor Comments:

In general for machine learning, an input data and expected results (target) are given and a model is trained by using these inputs for training. While I can see this in material methods, I think the author can expand this part for clarity.

For instance, the input matrix that has been used for training will have gene names from the model organism, whose data have been used for training. As a result, using this may be challenging to be used for other model organisms. What does training here means? Does Chord/ChordP has a weight matrix that can be used on any new datasets, or here training is to use every new input data for learning or optimization? This part is not clear to me.

Suggestion:

Even on fresh samples, more than 5% of cells may be doublets. However, for frozen human samples collected from patients, single-nuclear RNA-Seq (snuc-Seq) is commonly used and getting high quality data, removing doublets and low quality cells is important and can be more challenging. As a result, I suggest the authors to test Chord/ChordP on available snuc-Seq datasets as well.

For instance; <https://www.ncbi.nlm.nih.gov/geo/query/acc.cgi?acc=GSE147528>, which is publicly available.

Dear reviewer,

High-throughput single-cell RNA sequencing is accompanied by doublet problems that disturb the downstream analysis. Several computational approaches have been developed to detect doublets, but most of these methods are not always satisfactory in different datasets. Our study was designed to unveil a new method, Chord, which is an accurate and stable solution to the doublet detection problem. The Chord workflow is composed of three main steps. (i) Generating training data after coarse removal of doublets using primary methods and generating artificial doublets from the filtered data. (ii) Generalized Boosted Regression Modeling (GBM) model fitting which integrates and weights the predictions of published doublet detection tools based on classification performance on the training data. (iii) Application of the trained GMB model to the original dataset to predict doublets.

According to the suggestions of reviewers and editors, we made the following changes to the article:

Optimizations of Chord workflow:

- 1) Replace the Adaboost algorithm with the more efficient GBM algorithm.
- 2) Remove build-in method doubletCells which has poor performance.
- 3) Optimize the combination of doublet detection methods to improve ChordP's performance.

Changes of analysis:

- 1) The results of the previous version of Chord are replaced by the new version results in figures and tables (fig1, fig2g, fig4, figs1, figs2, figs3, table1, table s3-s11).
- 2) Display average pAUC800 pAUC900, pAUC950 pAUC975, AUC and PR of doublet detection methods in benchmark datasets to show

ChordP's performance more comprehensive (fig 2d).

- 3) Change the filtering criterions of DEGs and use R package Seurat 4.0.2 for calculation. Change the evaluation content through TPR, TNR and accuracy to make the result display more clearly (fig 3a).
- 4) According to suggestions from reviewers. In part "Applying Chord to real-world scRNA-Seq data", among the 19 microfluidic lanes in this data set, 5 tumor samples with the highest expected doublet rate were used for processing. The same evaluation as our previous version of the manuscript are performed and consistent conclusions are obtained (fig 4).
- 5) Describe the features of these doublets which trend to be a cluster in lung cancer data (fig 4b, c, d).
- 6) Replace line chart with bar chart to show the changes in the total number of differentially expressed genes before and after doublet removal (fig 4f).
- 7) Add cluster label to Figure4h in the trajectory analysis of lung cancer dataset, which visualize the trajectory changes at the cluster level (fig 4h).
- 8) According to suggestions from reviewers, calculate TP, FN, FP, TN, True Positive Rate(TPR), True Negative Rate(TNR), Precision, Accuracy in DM-A and HTO8 data sets (fig s2c).
- 9) Evaluate Chord's performance when using different boost algorithms and GBM is selected as Chord's default boost algorithms (fig s3a).
- 10) Test different combinations of methods for ChordP. In all combinations, "Chord+Scrublet+DoubletDetection" performs best (fig s3b).
- 11) Evaluate the robustness of the parameter "doubletrate" (fig s3e).

Writing:

- 1) Improve description of Chord workflow, especially how to train GBM models.
- 2) Describe usage details of the software cited in this article, such as SciBet and ROGUE.

- 3) In the method section, improve the description of execution steps, such as analysis on lung cancer data and the process of DEGs analysis.
- 4) Improve the description of assessment indexes, such as TPR, TNR and pAUC.
- 5) Correct grammatical errors and inaccurate descriptions.

Reviewer #1 (Remarks to the Author):

The authors describe Chord – a computational pipeline for scRNA-seq doublet detection that leverages an ensemble approach to aggregate the results of existing doublet detection tools. The authors begin by providing an overview of the Chord workflow, which includes three key steps: (i) Generation of training data after coarse removal of doublets using existing methods and generation of artificial doublets from the “cleaned” data, (ii) Adaboost model fitting which integrates and weights the predictions of existing doublet detection tools based on classification performance on the training data, and (iii) Application of the trained adaboost model to the original dataset to predict real doublets. We find the authors description of the Chord workflow to be well-considered and a notable contribution to the single-cell genomics doublet detection literature. However, we have concerns regarding the choice of doublet detection methods used during training data

generation (Major Comment #1) and adaboost model training (Major Comment #2), as well as concerns regarding documentation of scRNA-seq pre-processing and doublet method parameter fitting (Major Comment #3).

After describing the Chord workflow, the authors benchmark Chord against the four ‘built-in’ doublet detection methods used to generate the training data (e.g., DoubletFinder, doubletCells, bcde, and cxd) on two publicly-available PBMC scRNA-seq datasets where doublets are empirically-defined using cell hashing or in silico genotyping. This

analysis reveals that Chord performs with similar accuracy and stability to existing doublet tools on the full datasets, as well across datasets where doublets are proportionally up- and down-sampled. The authors then repeat this benchmarking workflow for Chord-plus (ChordP), which incorporates additional doublet detection methods (e.g., DoubletDetection, Solo, and Scrublet) into the Chord adaboost training step. This analysis reveals that including more doublet detection methods into the adaboost training step improves ChordP performance relative to Chord, although improvements over existing doublet detection methods remain modest. Altogether, we found these benchmarking workflows to be well-executed.

Having established that Chord and ChordP are amenable to scRNA-seq doublet detection, the authors then use simulated scRNA-seq datasets from Xi & Li (Cell Systems, 2020) to benchmark their method during pseudotemporal ordering and differential gene expression analyses (computational tasks known to be improved after doublet removal). This work reveals that Chord performs comparably to other doublet detection tools, although we disagree strongly with some of the claims the authors make in this section of the manuscript (Major Comment #4).

Finally, the authors apply Chord to 'real-world' scRNA-seq data of primary lung tissue from Lambrechts et al (Nature Medicine, 2018) where doublets are unknown, and demonstrate how Chord improves scRNA-seq analysis workflows downstream of doublet removal such as cell type purity (ROGUE), cell type identification (SciBet), differential gene expression analysis, and pseudotemporal ordering. While we appreciate the purpose of this analysis, we have serious concerns about how to authors applied Chord to these data (Major Comment #5) and believe that the authors should repeat this workflow on a scRNA-seq dataset that is more amenable to their purposes.

On balance, we consider the theoretical underpinnings of Chord to be sound and sufficiently novel, and find that the authors' benchmarking workflows were well-executed. Moreover, while Chord does not perform

better than existing doublet detection methods with regards to absolute accuracy or improvements in scRNA-seq analyses downstream of doublet removal, we are convinced by the authors' claims that Chord will perform more robustly across diverse scRNA-seq datasets due to its aggregated nature. For these reasons, we believe this manuscript should be conditionally-accepted for publication in Communications Biology, assuming that our Major Comments are adequately addressed.

Major Comments1

In both Chord and ChordP, the authors use "representative" computational doublet detection methods to remove doublets during the training data generation step. Specifically, they use the kNN classifiers DoubletFinder and doubletCells, as well as the marker gene co-expression classifiers bcds and cxds. While it is true that these are two distinct types of doublet detection tools, there are other doublet detection tools that could also be incorporated to make this step truly "representative". For example, Solo and DoubletDetection use neural networks and the hypergeometric test to classify doublets in a fashion that is distinct from bcds, cxds, DoubletFinder and doubletCells. Moreover, it is clear from the authors' own analyses that doubletCells performs quite poorly across most of the benchmarking datasets, which begs the question why doubletCells is relied upon at all for training data generation.

One critical analysis that we encourage the authors to include in the revised manuscript is a systematic analysis of Chord performance (e.g., mean AUC across benchmarking datasets) using all combinations of doublet detection methods for training data generation. Is there an ideal combination of methods to use for this step? Are there methods that decrease performance when they are included?

Reply 1 :

We removed doubletCells from Chord and ChordP. After evaluating 7 combinations (table r1) by calculating the specificity and sensitivity of

the generated data, we found that the suitability of overkill varies greatly on different samples (fig r1.1). In addition, we found that false positive doublets may enrich in some region in UMAP plot (fig r1.2). We were concerned that this might have a negative impact on simulation of Doublets. Based on the results, we use bcds, cxds and DoubletFinder as the default combination, and open the parameter selections of "overkill", so that advanced users can freely choose the method required by the overkill.

The assessment of overkill :

We evaluated 7 combinations in the step of overkill from different methods:

Name of combination	The methods in combination
Overkill-1	DoubletFinder,cxds
Overkill-2	bcds,DoubletFinder
Overkill-3	bcds,cxds
Overkill-4	bcds,cxds,DoubletFinder
Overkill-5	bcds,cxds,DoubletFinder,Solo
Overkill-6	bcds,cxds,DoubletFinder,Solo,DoubletDetection
Overkill-7	bcds,cxds,DoubletFinder,Solo,DoubletDetection,Scrublet

(table r1)

Overkill-5, Overkill-6 and Overkill-7 are based on the combination of bcds, cxds and doubletfinder by adding Solo, DoubletDetection and Scrublet. Because the average AUC of Solo, DoubletDetection and Scrublet in the test datasets is from high to low.

(fig r1.1)

Among the seven combinations, overkill-1 performed the worst in sensitivity, and overkill-2 and overkill-3 had their own advantages and disadvantages in different data sets. After gradually adding the number of combination methods, the accuracy will be improved, but the specificity will be reduced accordingly. We found that the specificity of the data set with a low expected doublet rate can still be maintained above 90% under the condition of high sensitivity, such as DM-A, DM-B and DM-C, and the data set with a high expected doublet rate, for example, HTO8 and HTO12, as the number of combined methods increase, the sensitivity increases, but its specificity is greatly reduced. When HTO12 uses 6 methods in the step of overkill, its specificity is only 51.3%. In addition, displaying false-positive doublets in the UMAP plot shows that adding too many methods will result in a large number of false positives and the formation of obvious clusters. Different clusters may be different cell types. Removing these false positives might affect the synthesis of simulated doublets from these cell types. Thereby this will introduce bias into the simulated doublet dataset.

HTO8 overkill-4 UMAP

HTO8 overkill-5 UMAP

HTO8 overkill6 UMAP

HTO8 overkill7 UMAP

(fig r1.2)

Major Comments2

In a similar vein as Major Comment #1, we are curious whether further optimization can be introduced with regards to which models are included in the adaboost training step. We see that ChordP improves upon Chord as more models are used – however, does this improvement represent the optimal performance of the method? Moreover, does inclusion of more models with more similar underlying algorithms bias the aggregated Chord result (in a fashion that reflects biases amongst the existing doublet detection literature)?

We encourage the authors to repeat the analysis suggested in Major Comment #1 but on the adaboost training step to identify whether Chord performance is sensitive to which sets of models are included.

Reply2 :

Based on your suggestions, we optimized the step of ensemble doublet detected methods. However, before evaluating the performance of different method combinations, we made another optimization. Firstly, we tested the performance of four ensemble algorithms, GBM, xgboost, lightgbm, and adaboost. The results show that the selection of the GBM algorithm has a significant improvement in the integration effect of the doublet scoring values (t.test, alternative = "greater", paired=T). Therefore, we modified gbm as the default ensemble method and re-evaluated the follow-up evaluation.

(fig s3a)

AUC:

	gbm	adaboost	xgboost	lightgbm
DM-A	0.828709094	0.823839679	0.815055328	0.783883994
DM-B	0.776636192	0.741326187	0.758812526	0.719949559
DM-C	0.831603921	0.826640536	0.820030036	0.776291885
DM-2.1	0.896349756	0.844874912	0.85716306	0.681049952

DM-2.2	0.896874132	0.835183151	0.854743351	0.692092863
HTO8	0.83297564	0.82238169	0.818058764	0.788461959
HTO12	0.619319805	0.608800927	0.612047955	0.60849529

(table r2.1)

PR:

	gbm	adaboost	xgboost	lightgbm
DM-A	0.437387219	0.422423682	0.378218451	0.123876233
DM-B	0.268571408	0.268511582	0.228284954	0.086083067
DM-C	0.463697516	0.436685827	0.416613875	0.164943788
DM-2.1	0.609248075	0.495271883	0.542787839	0.171111164
DM-2.2	0.627032041	0.495042412	0.542548	0.181290533
HTO8	0.622647098	0.59527651	0.585503212	0.382684894
HTO12	0.411325584	0.402126599	0.403308516	0.353428775

(table r2.2)

Then, after excluding doubletCells, different combinations were evaluated in HTO8, HTO12, DM-A, DM-B, DM-C, DM-2.1, and DM-2.2. We refer to the best performing combination as ChordPlus (ChordP). We will conduct corresponding evaluations on new methods that officially published in the future, and recommend a combination of methods in our github for users' reference.

(fig s3b)

The new version of ChordP has an average AUC of 0.8132 in the test data set, higher than the second-place SOLO's 0.8031. (Line:172-174)

Major Comments3

The authors note in the Methods section that Chord uses default parameter settings for the aggregated doublet detection models. However, it is key to note that some doublet detection models require parameter optimization for effective performance. For example, the authors implement DoubletFinder "with parameter selection PCs = 1:10, pN = 0.25, and automatically extracting the pk value corresponding to the highest bimodal coefficient to extract the doublet score." It must be noted that DoubletFinder is implemented to interface with a pre-processed Seurat object wherein the ideal number of PCs are determined by the user for their particular dataset before using the method. That is, hard-setting PCs = 1:10 will necessarily result in sub-optimal DoubletFinder (and, as a result, Chord) performance.

Moreover, it is not uncommon for mean-variance-normalized bimodality coefficient distributions to exhibit multimodality (this often happens when data is not properly quality-controlled), which in turn requires DoubletFinder users to manually interrogate the predictions after using distinct pK values to identify the "correct" parameters. As a result, automatically setting pK to the maximum bimodality coefficient value may further cause Chord performance to deteriorate.

Thus, we think it is very important that the authors build out the pre-processing guidelines for scRNA-seq data prior to using Chord, as well as parameter optimization guidelines for each doublet detection method used in their aggregated pipeline.

Reply3 :

We adopted the default parameters of each integrated software as the preset parameters in Chord. For example, for DoubletFinder, the parameter settings recommended by the original author are used, and according to the author's guide, the theoretical optimal PK through the find.pK() function (<https://github.com/chris-mcginnis-ucsf/DoubletFinder/>) are calculated. Indeed, we also think that may result in the inability to calculate the appropriate PK value, so we summarize the hyperparameter setting guide for users and set up the build-in methods' key parameter adjustable for users.

Major Comments4

While we found the differential gene expression analyses and pseudotemporal ordering comparisons to be well-executed, the authors posit a perplexing explanation for the increase in detected DEGs relative to "clean" synthetic data following doublet removal by some doublet detection methods. Specifically, the authors state in lines 194-197: "...TPRs of the datasets processed by some doublet detection methods were higher than those for the clean data, which may be due to the deletion of transition state cells between cell types that were identified as doublets through these methods, resulting in obvious differences

between cell types to detect more DEGs.”

This claim is confusing because the synthetic data used in Xi & Li (and this paper) does not include any “transition state cells”, as is evident in Fig. 3D of this manuscript. Please provide an explanation as to why we are incorrect in this interpretation, or remove this claim from the manuscript. Moreover, the authors should cite Xi & Li for the description of why TNR is high across all datasets.

Reply4 :

According to your valuable suggestion, we deleted the conclusion of improper description in the paper and quote Xi & Li's description of TNR. In previous assessment, we used Seurat 3.1.3's FindMarkers function to calculate DEGs, but in later versions the authors of Seurat made a series of changes to the FindMarkers function (<https://satijalab.org/seurat/news/index.html>). We updated the DEGs filtering standard and the software version used for calculation (Seurat 4.0.4), and adopted TNR, TPR and accuracy as evaluation indexes. (Line:243-244)

Major Comments5

The authors use the primary lung scRNA-seq data from Lambrechts et al to demonstrate the utility of applying Chord to ‘real-world’ scRNA-seq data. According to the associated Nature Medicine manuscript, this dataset was generated by running 19 distinct 10x Genomics microfluidic lanes representing normal lung tissue and tumor core/middle/edge tissue from 5 distinct patients, each with unique cellular compositions and cell loading densities. The explanation of how this data was analyzed is a bit sparse – so apologies if we are interpreting your work incorrectly – but was Chord run on the aggregated dataset including all of these samples? Or was Chord run independently on each sample prior to combining each sample? If it is the former, then Chord will be attempting to find doublets that could not possibly exist (e.g., tumor-tumor doublets from distinct patients; tumor-endothelial doublets from normal and

tumor samples, etc.) which would seriously confound these results. At a minimum, the authors must clarify how exactly they did this analysis – although other scRNA-seq datasets are likely to be more amenable to the points the authors intend to make (which we are confident will hold up).

Reply5 :

Thank you for pointing out this problem. We used Chord to re-evaluate the primary lung cancer scRNA-seq data set from Lambrechts et al. Among the 19 microfluidic lanes in this data set, 5 tumor samples with the highest expected doublet rate were used for processing. The same evaluation as our previous version of the manuscript were performed, and the conclusions obtained were basically the same as those in the previous version of the manuscript. (line:288-290)

Minor Comments1

1. Paper needs significant editing for clarity and grammar.

Reply1 :

Thank you for comments. We checked and revised the grammatical errors in the article.

Minor Comments2

The authors should explain more clearly why removing doublets during training data generation is beneficial for Chord performance. Also, the authors should clarify the workflow for generating artificial doublets – specifically, we are confused how weighting expression profiles from randomly-selected cells with a "biological random number from a $N(0.1,1)$ distribution" avoids generating synthetic doublets derived from the same cell type. Wouldn't it be easier to manually sample cell pairs from different gene expression clusters? What is the purpose of the random number generation?

Reply2 :

We added descriptions : “Doublets in the original data set might cause two types of potential errors which may be introduced into the training set: (i) In the process of generating doublets, the doublets will also be treated as singlets to simulate new doublets, resulting in wrong doublets are introduced into the training set. (ii) In the generated training set, the remaining doublets in the original data will be marked as singlets.”

(line 106-110)

When generating the simulated doublets, the random number of the normal distribution $N(0.1,1)$ is assigned to two single cells as mixing ratio, which could mimic the randomness of doublets generation in real experiments.

Minor Comments3

Lines 111-112: “Ground-truth comparisons illustrate that Chord detects most doublets” – The authors use qualitative comparisons of UMAP embeddings to make this quantitative claim. Authors should either provide the statistics underlying the claim or change the statement (e.g., Chord-derived doublet scores are enriched in regions of gene expression space associated with ground-truth doublet classifications).

Reply3 :

We analyzed the true positives and false positives of the doublet identification results using UMAP plot (fig S2a, b), and supplemented the corresponding statistical results (fig S2c). At the same time, we modified the description " Chord-derived doublet scores are enriched in regions of gene expression space associated with ground-truth doublet classifications".

DM-A								HTO8									
52	68	68	3110	0.4333	0.9786	0.4333	0.9588	bcds	1349	1249	1249	12723	0.5192	0.9106	0.5192	0.8492	bcds
46	74	74	3104	0.3833	0.9767	0.3833	0.9551	cxds	1369	1229	1229	12743	0.5269	0.912	0.5269	0.8517	cxds
56	64	64	3114	0.4667	0.9799	0.4667	0.9612	Chord	1596	1002	1002	12970	0.6143	0.9283	0.6143	0.8791	Chord
55	65	65	3113	0.4583	0.9795	0.4583	0.9606	ChordP	1642	956	956	13016	0.632	0.9316	0.632	0.8846	ChordP
11	109	109	3069	0.0917	0.9657	0.0917	0.9333	doubletCells	264	2334	2334	11638	0.1016	0.833	0.1016	0.7183	doubletCells
51	69	69	3109	0.425	0.9783	0.425	0.9582	DoubletFinder	1349	1249	1249	12723	0.5192	0.9106	0.5192	0.8492	DoubletFinder
45	75	75	3103	0.375	0.9764	0.375	0.9545	DoubletDetection	1594	1004	1004	12968	0.6135	0.9281	0.6135	0.8788	DoubletDetection
43	77	77	3101	0.3583	0.9758	0.3583	0.9533	Scrublet	1291	1307	1307	12665	0.4969	0.9065	0.4969	0.8422	Scrublet
32	88	88	3090	0.2667	0.9723	0.2667	0.9466	Solo	1712	886	886	13086	0.659	0.9366	0.659	0.8931	Solo
TP	FN	FP	TN	TPR	TNR	Precision	Accuracy		TP	FN	FP	TN	TPR	TNR	Precision	Accuracy	

(fig s2c)

Minor Comments4

Lines 568-569: Authors say they benchmarked Chord on the publicly-available HTO8 dataset which is comprised of "samples of cell lines using twelve barcoded antibodies to mark and label doublets." – According to the Seurat web-page and Stoeckius et al, HTO8 is scRNA-seq of PBMCs from 8 donors, while HTO12 is the dataset the authors are referring to in the Fig. 1 legend. The UMAP embedding looks like PBMCs, so the authors should change the text to reflect the correct dataset. The authors also need to change their references to HTO8 as "cell line" data throughout the manuscript and supplemental tables.

Reply4 :

It is a description mistake, and we corrected this error in the manuscript and supplemental tables.

Minor Comments5

Fig. S1A: "taning data set" instead of "training data set", "adboost" instead of "adaboost"

Reply5 :

We corrected this misspelling.

Minor Comments6

Lines 320-321: "...extracted the doublet scores evaluated by two

methods for each cell: i) through the DBF() function in the called R package DoubletFinder” – we are not aware of a 'DBF' function in DoubletFinder.

Reply6 :

DBF() is the function we defined in Chord. The function of DBF() is to run the DoubletFinder software and output the result. To avoid misunderstanding, we rewrote the description.

(Line412-415)

Dear reviewer,

High-throughput single-cell RNA sequencing is accompanied by doublet problems that disturb the downstream analysis. Several computational approaches have been developed to detect doublets, but most of these methods are not always satisfactory in different datasets. Our study was designed to unveil a new method, Chord, which is an accurate and stable solution to the doublet detection problem. The Chord workflow is composed of three main steps. (i) Generating training data after coarse removal of doublets using primary methods and generating artificial doublets from the filtered data. (ii) Generalized Boosted Regression Modeling (GBM) model fitting which integrates and weights the predictions of published doublet detection tools based on classification performance on the training data. (iii) Application of the trained GMB model to the original dataset to predict doublets.

According to the suggestions of reviewers and editors, we made the following changes to the article:

Optimizations of Chord workflow:

- 1) Replace the Adaboost algorithm with the more efficient GBM algorithm.
- 2) Remove build-in method doubletCells which has a mediocre performance.
- 3) Optimize the combination of doublet detection methods to improve ChordP's performance.

Changes of analysis:

- 1) The results of the previous version of Chord are replaced by the new version results in figures and tables (fig1, fig2g, fig4, figs1, figs2, figs3, table1, table s3-s11).
- 2) Display average pAUC800 pAUC900, pAUC950 pAUC975, AUC and

PR of doublet detection methods in benchmark datasets to show ChordP's performance more comprehensive (fig 2d).

- 3) Change the filtering criteria of DEGs and use R package Seurat 4.0.2 for calculation. Change the evaluation content through TPR, TNR and accuracy to make the result display more clearly (fig 3a).
- 4) According to suggestions from reviewers. In part "Applying Chord to real-world scRNA-Seq data", among the 19 microfluidic lanes in this data set, 5 tumor samples with the highest expected doublet rate were used for processing. The same evaluation as our previous version of the manuscript are performed and consistent conclusions are obtained (fig 4).
- 5) Describe the features of these doublets which trend to be a cluster in lung cancer data (fig 4b, c, d).
- 6) Replace line chart with bar chart to show the changes in the total number of differentially expressed genes before and after doublet removal (fig 4f).
- 7) Add figure labeled by clusters to Figure4h in the trajectory analysis of lung cancer dataset, which visualize the trajectory changes at the cluster level (fig 4h).
- 8) According to suggestions from reviewers, calculate TP, FN, FP, TN, True Positive Rate(TPR), True Negative Rate(TNR), Precision, Accuracy in DM-A and HTO8 data sets (fig s2c).
- 9) Evaluate Chord's performance when using different boost algorithms and GBM is selected as Chord's default boost algorithms (fig s3a).
- 10) Test different combinations of methods for ChordP. In all combinations, "Chord+Scrublet+DoubletDetection" performs best (fig s3b).
- 11) Evaluate the robustness of the parameter doubletrate (fig s3e).

Writing:

- 1) Improve description of Chord workflow, especially how to train GBM models.
- 2) Describe usage details of the software cited in this article, such as

SciBet and ROGUE.

- 3) In the method section, improve the description of execution steps, such as analysis on lung cancer data and the process of DEGs analysis.
- 4) Improve the description of assessment indexes, such as TPR, TNR and pAUC.
- 5) Correct grammatical errors and inaccurate descriptions.

Reviewer #2 (Remarks to the Author):

This manuscript proposes a new computational doublet-detection method based on the ensemble of multiple cutting-edge methods. The basic idea is 1) remove high-confidence doublets by four independent methods; 2) after removal, simulate artificial doublets by adding two randomly selected cells, and merge these artificial doublets back to obtain a training set; 3) train an Adaboost model on the training set, in which each independent method's doublet score vector is one predictor, and the singlet/artificial-doublet label from step one is the target variable; 4) apply the trained Adaboost to the original dataset for doublet prediction. The authors performed various validations on real and synthetic datasets to check the method's prediction accuracy and impacts on downstream analysis. The proposed ensemble framework is novel in the doublet-detection field. It also resolves some issues in other related studies. On the other hand, there are major concerns in the manuscript in terms of methods implementation, validation, and writing. I hope the authors could fix them to improve the current work.

Method and technical details1

In lines 307-309, the authors claimed that the performance of Chord is robust across different doublet rates yet without evidence. I suggest providing a sensitivity analysis to support this argument. Some other methods have similar parameters, for example, the number of artificial doublets in DoubletFinder 1 and Scrublet 2. The selection of those parameters has been clearly shown to be robust by sensitive analysis.

Reply1:

We added sensitivity analysis to show the robustness. Chord’s performance with different doubletrate from 0.11 to 0.29 on six data sets. These 6 datasets were randomly sampled and generated by the DEG test dataset, and the true doublet rate was 0.2. We didn't see a particularly noticeable deviation in AUC from the boxplot as the doubletrate parameter deviated from the true Doublet rate (0.2) (Figure S3e). At the same time, doubletrate in this range is not significant correlated with the average AUC value (p value=0.2792) (Table S11)

Figure S3e

	0.11	0.13	0.15	0.17	0.19	0.21	0.23	0.25	0.27	0.29	correlation(pearson)	p-value
data1	0.91237188	0.92059688	0.91972813	0.92207188	0.91439063	0.92786875	0.92953438	0.90994063	0.92065938	0.92229688	0.2243576	0.5332
data2	0.93799688	0.94195938	0.93335313	0.93415625	0.927625	0.93534063	0.93534063	0.92973125	0.9354625	0.93372813	-0.4188168	0.2283
data3	0.89879063	0.90381875	0.89544688	0.90135313	0.90796563	0.9075625	0.90040313	0.91124375	0.9095375	0.92343438	0.7774405	0.008125
data4	0.91879375	0.91471563	0.9063	0.915875	0.91372813	0.92125625	0.9260875	0.91120313	0.9067	0.85985	-0.5203941	0.1231
data5	0.93513125	0.94219375	0.93616875	0.93737813	0.93294063	0.93924063	0.9311375	0.93619688	0.92894688	0.9265	-0.7032218	0.02328
data6	0.92229063	0.93278438	0.93312188	0.93135625	0.9289875	0.92821563	0.92741563	0.9306625	0.92635625	0.92820938	-0.09368883	0.7968
mean AUC	0.92089583	0.92601146	0.92068646	0.92369844	0.92093958	0.92658073	0.92498646	0.92149635	0.92127708	0.91566979	-0.3796873	0.2792

Table S11

Method and technical details2

Line 326 introduces the overkillrate parameter and claims "we filtered the doubletrate*overkillrate percentage of cells ..." It seems that overkillrate is a numerical parameter that controls the conservativeness

of preliminary doublet removal. I hope the authors could provide practical guidance on how to choose this parameter and how robust the model performance is under different values. Although the software GitHub page sets this parameter a default value 1, the authors did not specify which value they use in the analysis of the manuscript.

Reply2:

In Chord and ChordP, the parameter overkillrate adopts the default value of 1, and this parameter is only set to increase the adjustability of the software. We use the combination "bcds+cxds+DoubletFinder" to test each data set under different overkillrates, where the overkillrate is increased from 1 to 2, and UMAP plot is used to visualize the results. The results showed that although more true positive doublets could be detected as the overkillrate increases, false positive doublets gradually increase and are enriched in some clusters. Therefore, we recommend that the overkillrate value adopts 1 in the overkill step, that mean to remove the Doublets identified by the expected doublet rate of bc ds, cx ds and DoubletFinder.

Method and technical details3

After reading the "Model training" part (line 345-351) initially, I did not understand how the Adaboost model was trained, until I checked the figure on the GitHub page. I suggest adding those figures into the manuscript and explicitly demonstrating that each method's doublet score serves as the predictor/variable in the Adaboost model.

Reply3:

Based on your suggestions, we rewrote this part. Besides, we also optimized the ensemble algorithms. We first tested the performance of four ensemble algorithms, GBM, xgboost, lightgbm, and Adaboost. The results show that the selection of the GBM algorithm has a significant improvement in the integration effect of the doublet scoring values (t.test, alternative = "greater", paired=T). Therefore, we modified GBM as the default ensemble method and then rewrote the description.

(fig S3a)

We update the specific description of the training to make it clearer and explicitly demonstrating that each method's doublet score serves as the predictor.

"The Chord workflow is composed of three main steps (Figure 1a). (i) Generating training data after coarse removal of doublets using existing methods and generating artificial doublets from the filtered data. (ii) Generalized Boosted Regression Modeling (GBM) model fitting which integrates and weights the predictions of existing doublet detection tools based on classification performance on the training data. (iii) Application of the trained GBM model to the original dataset to predict doublets."

(Line 100-105)

"After evaluating the simulation training set using DoubletFinder, bcdrs and cxds to get their predicted scores, the GBM algorithm was adopted to integrate these predicted scores which served as the predictors in the GBM model." (line 117-119)

Method and technical details4

The methods included in Chord need to be reconsidered. First, there is one method, scDbFinder, not in the ensemble or extension part. This method was published last year and has been shown excellent performance and speed compared with others. Second, the doubletCells method has been shown to significantly underperform others in the previous benchmark study, Figure 2, and Figure S1. Actually, doubletCells was removed from the latest version of package scran (v 1.20.1), and the author of scran recommends using scDbFinder to substitute doubletCells in that package. Therefore, I suggest using scDbFinder instead of doubletCells as one of the four backbones. At least scDbFinder should be included in the ChordP.

Reply4:

We removed doubletCells, and recombined the methods and evaluated the integration effects of adding different combinations of methods on the basis of Chord. Then we set the best-performing combination as ChordPlus. We will evaluate the new officially published method in the future, evaluate whether the method is suitable for integration, and recommend the optimal combination method on our github for users' reference. scDbIFinder is an ensemble method based on xgboost, which is not an individual method and not suitable to be included in our integration.

Figure S3b

Method and technical details5

In the "Cluster purity" part (line 440), the author should clarify which package and function were used to calculate the ROGUE value. While I can understand by checking the GitHub page of ROGUE, it is not user-friendly without a clear illustration. The same issue applies to SciBet method, where the execution details were not shown.

Reply5:

We added descriptions of ROGUE and SciBet, as well as implementation details.

“Secondly, we used SciBet, a cell type annotation tool based on Bayes decision, to annotate the cell types from the original data and filtered data, and calculated the changes in the cell types before and after applying the doublet filter.” (line:306-308)

“We utilized ROGUE an entropy-based metric for assessing the purity of cell types in the original data and filtered data. The ROGUE index has been scaled to the range of zero to one where the larger value means the higher purity.” (line:302-304)

“Automated cell type annotation of this data set was performed using SciBet. The training model provided by SciBet, which was trained from 42 human single-cell datasets containing 30 major human immune cell types, was used to automatically annotate the datasets before and after doublet removal.” (fig S3d)

Validation1

The comparison of detection accuracy on real data is not convincing. In Table S3, Chord or ChordP only achieved two highest AUROCs and one highest AUPRC on seven real datasets. The proposed methods are not attractive if they did not show a strong advantage over single methods. This is a major concern also because the running time of Chord or ChordP will be the sum of all methods included plus the training time of Adaboost. One potential solution, as stated in the last section, is to substitute doubletCells with scDbIFinder or another better single method.

Reply1:

We removed doubletCells from the overkill step and the model integration step. At the same time, we adjusted the algorithm used for integration to Gradient Boosting Machine (GBM). After adjustment, the performance of ChordP and Chord has been improved. In the 7 data sets tested, the average AUC of ChordP increased from 0.806 to 0.813, the average PR increased from 0.427 to 0.467, the average AUC of Chord

increased from 0.793 to 0.801, and the average PR increased from 0.395 to 0.465.

Validation2

One of the main reasons for detecting doublets is to remove spurious clusters in the datasets. The importance of this goal has been demonstrated repeatedly in previous studies 1,2,5. While the authors evaluated the impacts of doublet-detection on the quality of true clusters, the capacity of removing spurious clusters has not been discussed. I suggest adding this analysis, either through purely simulated data or real data with artificial doublets.

Reply2:

We totally agree that the capacity of removing spurious clusters is valuable to be discussed. Therefore, we tried to find spurious clusters by increasing resolution (resolution =1.2) in the clustering step (function FindClusters() of R package Seurat 4.0.2). However, we found that even if we used a high resolution (default resolution=0.8), it was difficult to find a spurious cluster where most cells were doublets. Doublets might not easily cluster into separate spurious clusters, but rather enrich in partial clusters. In addition, the results of clustering may depend on resolution and other parameters. Based on these, we think that the capacity of removing spurious clusters may be difficult to accurately evaluate.

DM-A (doublet rate 0.036)

HTO8 (doublet rate 0.157)

Validation3

In practice, users will choose a cutoff on doublet scores to call doublets. I suggest adding one validation, which is to calculate the precision, recall, and true negative rate under certain doublet rates on real data. This analysis can give a more comprehensive evaluation of proposed methods.

Reply3:

We added these analysis with the HTO8 and DM-A data sets. We calculated the case of the doublet rate as a threshold, and calculated the accuracy, recall rate and true rate. The overall performance of these indicators is slightly different from that of AUC, but the trends are consistent.

DM-A								
52	68	68	3110	0.4333	0.9786	0.4333	0.9588	bcds
46	74	74	3104	0.3833	0.9767	0.3833	0.9551	cxds
56	64	64	3114	0.4667	0.9799	0.4667	0.9612	Chord
55	65	65	3113	0.4583	0.9795	0.4583	0.9606	ChordP
11	109	109	3069	0.0917	0.9657	0.0917	0.9339	doubletCells
51	69	69	3109	0.425	0.9783	0.425	0.9582	DoubletFinder
45	75	75	3103	0.375	0.9764	0.375	0.9545	DoubletDetection
43	77	77	3101	0.3583	0.9758	0.3583	0.9533	Scrublet
32	88	88	3090	0.2667	0.9723	0.2667	0.9466	Solo
TP	FN	FP	TN	TPR	TNR	Precision	Accuracy	

HTO8								
1349	1249	1249	12723	0.5192	0.9106	0.5192	0.8492	bcds
1369	1229	1229	12743	0.5269	0.912	0.5269	0.8517	cxds
1596	1002	1002	12970	0.6143	0.9283	0.6143	0.8791	Chord
1642	956	956	13016	0.632	0.9316	0.632	0.8846	ChordP
264	2334	2334	11638	0.1018	0.833	0.1018	0.7183	doubletCells
1349	1249	1249	12723	0.5192	0.9106	0.5192	0.8492	DoubletFinder
1594	1004	1004	12968	0.6135	0.9281	0.6135	0.8788	DoubletDetection
1291	1307	1307	12665	0.4969	0.9065	0.4969	0.8422	Scrublet
1712	886	886	13086	0.659	0.9366	0.659	0.8931	Solo
TP	FN	FP	TN	TPR	TNR	Precision	Accuracy	

(fig S2c)

Writing

Some sections of this manuscript were either not written in the academic style or with confusion. Some examples include:

- 1. There are several "STAR Methods" in the manuscript, which seems a unique item in articles from Cell Press.*
- 2. Line 122 "we used random sampling to proportionally sample singlets..." seems redundant expression.*
- 3. Line 152 pAUCXXX was shown without definition. This may confuse readers who lack expertise in machine learning terminology.*
- 4. Line 249 "A pseudotime analysis of the myeloid cells was conducted". Please consider rewriting this sentence.*
- 5. Line 348 "The R package 'adabag', through the AdaBoost algorithm, DBboostTrain() is used to implement model training for training based on the scoring results of the four methods." This sentence is ambiguous. Please consider rewriting.*
- 6. Line 370 "Doublet detection" should be DoubletDetection.*
- 7. Line 421 "the highest scoring cells for the number of double cells according to the results were removed". Consider rewriting this sentence.*
- 8. Line 456 should be followed by lines 458 and 459.*

Reply:

Thanks for pointing out these problems. We corrected the above errors and similar errors in the manuscript by referring to the suggestions above and those of other reviewers.

1. Change to "Methods"

2. line 135-136

"we used random sampled singlets and doublets in the dataset to build a doublet rate gradient"

3. line 182-184

"Moreover, partial areas under the ROC curve (pAUC) at 80%(pAUC800), 90%(pAUC900), 95%(pAUC950) and 97.5%(pAUC975) specificity was calculated"

4. line316-318

"In the pseudotime of myeloid cells in the dataset, the doublets were unevenly distributed in the dimensionality reduction plot and aggregated on the right side."

5. line438-440

"GBM (R package gbm) which performed better than adaboost, xgboost, and lightgbm (Figure S3a) was used to combine the prediction scores of the build-in methods to fit a model for robust estimate."

6. line 464

"DoubletDetection settings"

7. line 516-518

"After evaluating the contaminated data set through a doublet detection method, cells predicted to be doublets were deleted."

8. The table of software was removed from the manuscript after revise.

Dear reviewer,

High-throughput single-cell RNA sequencing is accompanied by doublet problems that disturb the downstream analysis. Several computational approaches have been developed to detect doublets, but most of these methods are not always satisfactory in different datasets. Our study was designed to unveil a new method, Chord, which is an accurate and stable solution to the doublet detection problem. The Chord workflow is composed of three main steps. (i) Generating training data after coarse removal of doublets using primary methods and generating artificial doublets from the filtered data. (ii) Generalized Boosted Regression Modeling (GBM) model fitting which integrates and weights the predictions of published doublet detection tools based on classification performance on the training data. (iii) Application of the trained GMB model to the original dataset to predict doublets.

According to the suggestions of reviewers and editors, we made the following changes to the article:

Optimizations of Chord workflow:

- 1) Replace the Adaboost algorithm with the more efficient GBM algorithm.
- 2) Remove build-in method doubletCells which has poor performance.
- 3) Optimize the combination of doublet detection methods to improve ChordP's performance.

Changes of analysis:

- 1) The results of the previous version of Chord are replaced by the new version results in figures and tables (fig1, fig2g, fig4, figs1, figs2, figs3, table1, table s3-s11).
- 2) Display average pAUC800 pAUC900, pAUC950 pAUC975, AUC and PR of doublet detection methods in benchmark datasets to show ChordP's performance more comprehensive (fig 2d).

- 3) Change the filtering criteria of DEGs and use R package Seurat 4.0.2 for calculation. Change the evaluation content through TPR, TNR and accuracy to make the result display more clearly (fig 3a).
- 4) According to suggestions from reviewers. In part "Applying Chord to real-world scRNA-Seq data", among the 19 microfluidic lanes in this data set, 5 tumor samples with the highest expected doublet rate were used for processing. The same evaluation as our previous version of the manuscript are performed and consistent conclusions are obtained (fig 4).
- 5) Describe the features of these doublets which trend to be a cluster in lung cancer data (fig 4b, c, d).
- 6) Replace line chart with bar chart to show the changes in the total number of differentially expressed genes before and after doublet removal (fig 4f).
- 7) Add cluster label to Figure4h in the trajectory analysis of lung cancer dataset, which visualize the trajectory changes at the cluster level (fig 4h).
- 8) According to suggestions from reviewers, calculate TP, FN, FP, TN, True Positive Rate(TPR), True Negative Rate(TNR), Precision, Accuracy in DM-A and HTO8 data sets (fig s2c).
- 9) Evaluate Chord's performance when using different boost algorithms and GBM is selected as Chord's default boost algorithms (fig s3a).
- 10) Test different combinations of methods for ChordP. In all combinations, "Chord+Scrublet+DoubletDetection" performs best (fig s3b).
- 11) Evaluate the robustness of the parameter "doubletrate" (fig s3e).

Writing:

- 1) Improve description of Chord workflow, especially how to train GBM models.
- 2) Describe usage details of the software cited in this article, such as SciBet and ROGUE.
- 3) In the method section, improve the description of execution steps,

such as analysis on lung cancer data and the process of DEGs analysis.

- 4) Improve the description of assessment indexes, such as TPR, TNR and pAUC.
- 5) Correct grammatical errors and inaccurate descriptions.

Reviewer #3 (Remarks to the Author):

The manuscript, "Chord: Identifying Doublets in Single-Cell RNA Sequencing Data by an Ensemble Machine Learning Algorithm" by Xiong et al. has focused doublet detection in single-cell Sequencing datasets. Doublets/Multiplets in single-cell datasets are one of the challenging problems in field of single-cell droplet sequencing and detecting and removing them from the analyses, can have significant effect on down-stream analyses, depending on removing False Positive or False Negative Doublets. As a result, tools that can detect these doublets/multiplets in single-cell RNA-Seq are important and will be beneficial in the field. The authors have developed a tool named Chord and ChordP, which detect the doublets with high accuracy. Especially, combining the power of many available tools and machine learning are the two important points that I think makes this an excellent tool in the field and this manuscript should be considered for publication. I have some minor comments which are not clear in the manuscript and need clarification (at least for me).

Minor Comments:

In general for machine learning, an input data and expected results (target) are given and a model is trained by using these inputs for training. While I can see this in material methods, I think the author can expand this part for clarity.

For instance, the input matrix that has been used for training will have gene names from the model organism, whose data have been used for training. As a result, using this may be challenging to be used for other model organisms. What does training here means? Does Chord/ChordP has a weight matrix that can be used on any new datasets, or here training is to use every new input data for learning or optimization? This part is not clear to me.

Reply:

We are sorry for the unclear description of the manuscript. We evaluated and replaced a more effective integration algorithm GBM instead of Adaboost, and we updated the description of the integration step process in the text and methods section, so that it can describe our method more accurately.

Our model does not use a fixed data set to generate a model, and then apply the fixed model to other data. Instead, it automatically generates a training set based on the data itself and performs model training based on it. Then get the most suitable model for the data, so Chord/ChordP can also work normally on new input data.

The training step of GBM:

“After evaluating the simulation training set using DoubletFinder, bcdrs and cxds to get their predicted scores, the GBM algorithm was adopted to integrate these predicted scores which served as the predictors in the GBM model. Then the doublet scores output was calculated by the GBM model for the input droplets data (Figure S1a)” (Line: 116-120)

“GBM (R package gbm) which performed better than adaboost, xgboost, and lightgbm (Figure S3a) was used to combine the prediction scores of the build-in methods to fit a model for robust estimate. In GBM, each individual model consists of classification or regression trees, also called boosted regression trees (BRT). We defined 1000 trees for fitting, and set parameter shrinkage = 0.01, cv.folds = 5. Function DBboostTrain() was defined to implement model training, which combined the scoring results of these build-in methods into a matrix. Then the matrix was

input data for the function `gbm()` in R package `gbm`. The simulated doublets were set as true positives (TPs), and the singlets were set as true negatives (TNs) for model training.” (Line: 438-446)

Suggestion:

Even on fresh samples, more than 5% of cells may be doublets. However, for frozen human samples collected from patients, single-nuclear RNA-Seq (snuc-Seq) is commonly used and getting high quality data, removing doublets and low quality cells is important and can be more challenging. As a result, I suggest the authors to test Chord/ChordP on available snuc-Seq datasets as well.

For

instance; <https://www.ncbi.nlm.nih.gov/geo/query/acc.cgi?acc=GSE147528>, which is publicly available.

Reply:

It's meaningful to test Chord/ChordP on snuc-Seq datasets. For the data recommended, we selected "GSM4432635_SFG2", "GSM4432642_SFG9" and "GSM4432643_SFG8" which have the largest cell number for testing.

GSM4432635_SFG2

GSM4432642_SFG9

GSM4432643_SFG8

In the GSM4432635_SFG2 data, we simulated the doublets by adding the expression matrices of two random cells and adding them to the data. Chord clearly identified most areas where simulated doublets are concentrated.

However, currently we have not found the single-cell nuclear sequencing data with doubles labeling through experiments, so we are unable to comprehensively evaluate Chord on snuc-Seq. We will continue to pay

attention to the relevant data in the follow-up and supplement rigorous evaluation.

Reviewers' comments:

Reviewer #1 (Remarks to the Author):

Major Comments

1. The authors did not satisfactorily address Major Comment #3, which discusses PC and pK selection for DoubletFinder, and continued to use the hard-set PC=1:10 argument for their method, which is incorrect. PC and pK needs to be chosen in a dataset-specific fashion. This could explain why the authors observe a concentration of DoubletFinder false-positives in UMAP space when applied to the HTO-8 dataset (lines 296-298), which was not observed in the original DoubletFinder paper where the method was benchmarked on the same HTO-8 dataset.

2. Since doubletCells is no longer being used as part of Chord, the authors need to adjust the text accordingly. For example:

- Remove reference in Fig. 1 legend
- Remove from panels in Fig. 2 to enable better comparisons between methods (doubletCells is an obvious outlier that make such comparisons difficult visually)
- Lines 905-906: Authors say in the "Preliminary deletion of doublets" section, "doubletCells() was used to evaluate doublet cells with parameters 906 $k = 50$, $d = 50$ and extract the scores." — the authors do not use doubletCells in this step (unless we are misunderstanding).

3. Unless we are misunderstanding, the authors did not satisfactorily address Minor Comment #2. From the Methods section, artificial doublets are generated using the following workflow: (1) Normalize data using Seurat, (2) PCA and unsupervised clustering to get 20 total clusters (side note: this feels like a strange heuristic to apply), (3) Proportionally-sample cells from each cluster according to doublet rate and weight cell gene expression counts according to random number from $N(1,0.1)$ distribution, (4) Average cell pair gene expression profiles using these weights, and (5) Add simulated doublets into filtered data to make the training set.

We agree with the authors that doublets are generated randomly in real experiments, but this concept applies to the process of cell encapsulation itself, not transcript capture/barcoding within the droplet. Once two cells are lysed in the droplet, transcripts from each cell are sampled according to the total RNA content of each component cell type, not according to a random number from an $N(1,0.1)$ distribution. It is unclear if the authors have tested whether/how this random sampling alters Chord performance, but this should be explored explicitly because the method assumption doesn't reflect reality.

4. Figure 4A is confusing — heterotypic doublets are from two cell types. How are the authors assigning doublets to single cell types? Authors do not describe this in the methods or anywhere in the manuscript.

5. For the lung cancer data, it remains unclear whether the authors applied Chord to the aggregated data from the 5 tumor samples from distinct patients (which would be incorrect) or applied Chord individually to each sample before aggregation (which would be correct, in order to avoid attempting to predict inter-patient doublets that could not exist in the real data).

6. Lines 480-483: "According to the number of predicted doublets in different clusters, cluster 10 had the highest doublets enrichment trend (Figure 4b, Figure 4c). The doublets in cluster 10 simultaneously expressed markers of T cells and plasma cells (Figure 4d)" — This is a confusing result because plasma cells (normally MZB1+ cells co-expressing B-cell markers such as MS4A1) are not shown in the previous figure panels are normally quite rare compared to their B-cell precursors. Authors need to refine this analysis — do they mean B-cells?

7. It is unclear why the authors apply pseudotemporal ordering to infer myeloid cell trajectories in the tumor data — is there a biological rationale for expecting myeloid cells to be undergoing a differentiation trajectory in this system?

Minor Comments

1. Lines 82-84: "However, there are inherent limitations to these experimental techniques used for doublet detection. First, these methods require their special experimental operations and additional costs, so they are not suitable for existing scRNA-seq data." — It is true that multiplexing approaches require extra sample handling/costs, but this is unrelated to their inability to be applied retroactively to existing scRNA-seq datasets.

2. Lines 157-159: "We called this step "overkill" and an adjustable parameter called "overkillrate" to preliminarily delete doublets. Selecting this parameter could improve the accuracy of the program (Methods; Figure S1a)." — Authors need to clarify this statement, especially since Fig. S1a is just a schematic of the Chord method and doesn't specifically address whether "overkillrate" parameter selection could improve accuracy.

3. Reviewer response #3 (although I don't totally agree with the reviewer that this is a necessary analysis): There are many papers currently in press that apply sample multiplexing approaches to nuclei, for example:

<https://www.sciencedirect.com/science/article/pii/S1934590921001557>

<https://www.nature.com/articles/s41467-019-10756-2>

<https://www.ncbi.nlm.nih.gov/pmc/articles/PMC7724892/>

4. Paper still needs significant editing for grammar/clarity.

Reviewer #2 (Remarks to the Author):

The authors addressed my concerns in the first-round review. Thank you!

Reviewer #3 (Remarks to the Author):

The current version of the manuscript by Xiong et al., has been improved based on the reviewer comments. The author addressed all of my questions.

The author has evaluated the power of Chord/P with the available single-cell datasets. I think, a follow-up and rigorous evaluation of the Chord/P will improve the power of doublet detection by Chord/P but not required for the current study for publication.

I recommend the paper to be accepted with the current version.

Dear reviewer,

We would like to start by thanking the thoughtful comments. We have considered the comments and addressed the concerns raised.

Major Comments1

The authors did not satisfactorily address Major Comment #3, which discusses PC and pK selection for DoubletFinder, and continued to use the hard-set PC=1:10 argument for their method, which is incorrect. PC and pK needs to be chosen in a dataset-specific fashion. This could explain why the authors observe a concentration of DoubletFinder false-positives in UMAP space when applied to the HTO-8 dataset (lines 296-298), which was not observed in the original DoubletFinder paper where the method was benchmarked on the same HTO-8 dataset.

Reply1 :

The reviewer touches upon an important issue and pointed out that a concentration of DoubletFinder false-positives we observed may be caused by unspecific selection of PC and pK. However, we cannot fully agree with the comment.

First of all, if we interpret correctly, the supplementary figures from the original DoubletFinder paper did not label false positives in the HTO-8 visualization (please see SFig1B of DoubletFinder paper), and it was not specified like we did. Besides, even in the case of "ground-truth", the sensitivity of HTO-8 was 0.64, indicating there were also false positives in their DoubletFinder's results. Therefore, both of our results showed false-positives, which may not be attributed to the incorrect parameters setting.

SFig1B of DoubletFinder paper

Second, the authors of DoubletFinder recommended PC parameter set as 1:10 (<https://github.com/chris-mcginnis-ucsf/DoubletFinder>) which was also directly used for all data sets in their study. (<https://doi.org/10.1016/j.cels.2019.03.003>)

Seurat Pre-processing Parameters							
Data	REF	PCs	Variable gene dispersion threshold	Variable gene expression threshold	pN	pK	# of doublet predictions
Demuxlet	Kang et al., 2018	10	0.85	0.05	0.25	0.01	6909 cells
Cell Hashing	Stoeckius et al., 2018	10	0.65	0.025	0.25	0.01	2687 cells
Kidney	Park et al., 2018	10	0.25	0.0125	0.25	0.09	913, 473 cells

Parameters of DoubletFinder paper

In addition, the choice of this parameter has been widely recognized, and the recommended PC=1:10 has been adopted in many papers involving the evaluation of DoubletFinder, for example:

<https://doi.org/10.1016/j.cels.2020.11.008>

<https://doi.org/10.1093/bioinformatics/btz698>

<https://doi.org/10.1016/j.celrep.2019.09.082>

PC1:10 might be a parameter with generality. For example, on the same HTO-8 dataset, function ElbowPlot() in Seurat was applied to calculate the standard deviations of the principle components. An elbow is observed around the 10 PC in the graph, which suggests that the majority of true signal is probably captured in the first 10 PCs.

Standard deviations of PCs (HTO8)

Standard deviations of PCs (DM-A)

Standard deviations of PCs (DM-B)

Standard deviations of PCs (DM-C)

Standard deviations of PCs (DM-2.1)

Standard deviations of PCs (DM-2.2)

Standard deviations of PCs (HTO12)

Finally, as for the pK value setting, the DoubletFinder paper explicitly states that “DoubletFinder parameters must be selected using a ground-truth-agnostic strategy called mean-variance-normalized bimodality coefficient (BCMVN) maximization”, and they suggested “Optimal pK for any scRNA-seq data can be manually discerned as maxima in BCmvn distributions.” in their instruction of function find.pK(). Therefore, we set pK values in this way. We think the reviewer’s point “it is not uncommon for mean-variance-normalized bimodality coefficient distributions to exhibit multimodality (this often happens when data is not properly quality-controlled)” is valuable, so we have set pK and PC in Chord as parameters that can be adjusted by users.

Major Comments2

Since doubletCells is no longer being used as part of Chord, the authors need to adjust the text accordingly. For example:

- *Remove reference in Fig. 1 legend*
- *Remove from panels in Fig. 2 to enable better comparisons between methods (doubletCells is an obvious outlier that make such comparisons difficult visually)*
- *Lines 905-906: Authors say in the “Preliminary deletion of doublets” section, “doubletCells() was used to evaluate doublet cells with parameters 906 $k = 50$, $d = 50$ and extract the scores.” — the authors do not use doubletCells in this step (unless we are misunderstanding).*

Reply2 :

We appreciate the reviewer's helpful suggestions and the careful review.

- We have adjusted the text about doubletCells accordingly.
- We hope to keep the panel of doubletCells, because this method is well recognized and the fact that it is an outlier explains why we did not integrate it into ChordP. Also, Heatmap in Fig 2 was re-plotted without doubletCells (please see the figure below). Judging by the results, with or without doubletCells would not affect comparisons particularly.

Fig 2b (comparisons were made without doubletCells)

- We have removed the redundant description.

Major Comments3 :

Unless we are misunderstanding, the authors did not satisfactorily address Minor Comment #2. From the Methods section, artificial doublets are generated using the following workflow: (1) Normalize data using Seurat, (2) PCA and unsupervised clustering to get 20 total clusters (side note: this feels like a strange heuristic to apply), (3) Proportionally-sample cells from each cluster according to doublet rate and weight cell gene expression counts according to random number

from $N(1,0.1)$ distribution, (4) Average cell pair gene expression profiles using these weights, and (5) Add simulated doublets into filtered data to make the training set.

We agree with the authors that doublets are generated randomly in real experiments, but this concept applies to the process of cell encapsulation itself, not transcript capture/barcoding within the droplet. Once two cells are lysed in the droplet, transcripts from each cell are sampled according to the total RNA content of each component cell type, not according to a random number from an $N(1,0.1)$ distribution. It is unclear if the authors have tested whether/how this random sampling alters Chord performance, but this should be explored explicitly because the method assumption doesn't reflect reality.

Reply3 :

In the process of doublets generation, mRNA degradation of two cells in the same droplet is different, and ambient RNA in background contamination may cause the cell mixing ratio to deviate from 1:1. Erica A.K. DePasquale, the author of *DoubletDecon: Deconvoluting Doublets from Single-Cell RNA-Sequencing Data*, also discussed about mixing cells ratio: "differing RNA abundance and/or technical variation in cDNA generation may result in uneven contribution from each cell. Hence, modeling doublets as an equal contribution of two different cells is likely to be overly simplistic". (<https://doi.org/10.1016/j.celrep.2019.09.082>) So, in our study, we added a random number from an $N(1,0.1)$ distribution to roughly represent these randomness. We apologize for the unclear description and we have made modifications in the manuscript accordingly.

Further discussion on the factors that potentially alter the cell mixing rate may be required another study to pursue, however, it falls out of the scope of the current study.

Major Comments4 :

Figure 4A is confusing — heterotypic doublets are from two cell types.

How are the authors assigning doublets to single cell types? Authors do not describe this in the methods or anywhere in the manuscript.

Reply4 :

We apologize for the unclear statement. We wanted to show doublets ratio of all the cell types which had already been labelled by the lung cancer paper. In that study, the author did not perform doublets detection, so we calculated the percentage of doublets in cells which assigned to single cell types by their study.

We amended the description to "After detecting doublets by Chord on the labelled cell from the original paper (<https://gbiomed.kuleuven.be/scRNAseq-NSCLC>)".

Line:285-286

Major Comments5 :

For the lung cancer data, it remains unclear whether the authors applied Chord to the aggregated data from the 5 tumor samples from distinct patients (which would be incorrect) or applied Chord individually to each sample before aggregation (which would be correct, in order to avoid attempting to predict inter-patient doublets that could not exist in the real data).

Reply5 :

We applied Chord individually to each sample and labelled doublets, then we integrated 5 tumor samples to aggregated data for other analysis. We add description in Method accordingly.

"After evaluating the predicted double cell rate for each sample based on the number of cells, we selected the 5 tumour samples (sample 11, 13, 17, 18, 22) with the highest predicted double cell rate and applied Chord individually to each sample. Then we performed a standard Seurat analysis"

line:540-543

Major Comments6 :

Lines 480-483: "According to the number of predicted doublets in different clusters, cluster 10 had the highest doublets enrichment trend (Figure 4b, Figure 4c). The doublets in cluster 10 simultaneously expressed markers of T cells and plasma cells (Figure 4d)" — This is a confusing result because plasma cells (normally MZB1+ cells co-expressing B-cell markers such as MS4A1) are not shown in the previous figure panels are normally quite rare compared to their B-cell precursors. Authors need to refine this analysis — do they mean B-cells?

Reply6 :

According to the cell type label of the original paper of this dataset, B cells were not further classified to cell subtype in this dataset. As shown in Figure 4b, B cells consist of 2 clusters (cluster 6 and 8), where cluster 8 is the plasma cell, proven by cell specific markers like XBP1, SSR4, SSR3, and CD38. We have added a description about this into manuscript: " Cluster 10 is shown to be the closest neighbor to both cluster 1 and cluster 8 on the UMAP plot (Figure 4b). Cluster 1 is T cell cluster, while cluster 8 is plasma cell cluster which is a cell subtype of B cells. "

Line:293-296

Major Comments7 :

It is unclear why the authors apply pseudotemporal ordering to infer myeloid cell trajectories in the tumor data — is there a biological rationale for expecting myeloid cells to be undergoing a differentiation trajectory in this system?

Reply7 :

Accurate prediction of the cell trajectory would benefit interpretation of tumor data in real situation. We wanted to demonstrate methodologically that Chord is able to correct the direction of the cell trajectory. Since myeloid has the highest proportion of doublets (8.15%)

and have a biological rationale in tumour microenvironment (<https://doi.org/10.1016/j.cell.2021.01.010>). So, myeloid cells were selected for analysis. We have changed our description in the manuscript in lines 314-320.

Minor Comments1 :

Lines 82-84: "However, there are inherent limitations to these experimental techniques used for doublet detection. First, these methods require their special experimental operations and additional costs, so they are not suitable for existing scRNA-seq data." — It is true that multiplexing approaches require extra sample handling/costs, but this is unrelated to their inability to be applied retroactively to existing scRNA-seq datasets.

Reply1 :

We changed it to "First, these methods require their special experimental operations and additional costs. Moreover, these experimental techniques are not suitable for existing scRNA-seq data."

Line:62-64

Minor Comments2 :

Lines 157-159: "We called this step "overkill" and an adjustable parameter called "overkillrate" to preliminarily delete doublets. Selecting this parameter could improve the accuracy of the program (Methods; Figure S1a)." — Authors need to clarify this statement, especially since Fig. S1a is just a schematic of the Chord method and doesn't specifically address whether "overkillrate" parameter selection could improve accuracy.

Reply2 :

We modified the description of "overkillrate".

"Selecting "overkill" could improve the accuracy of training sets which is beneficial to model fitting (Methods; Figure S1c)."

line:112-113

Minor Comments3 :

Reviewer response #3 (although I don't totally agree with the reviewer that this is a necessary analysis): There are many papers currently in press that apply sample multiplexing approaches to nuclei, for example:

<https://www.sciencedirect.com/science/article/pii/S193459092100155>

Z

<https://www.nature.com/articles/s41467-019-10756-2>

<https://www.ncbi.nlm.nih.gov/pmc/articles/PMC7724892/>

Reply3 :

We thank the reviewer for the helpful information.

Minor Comments4 :

Paper still needs significant editing for grammar/clarity.

Reply4 :

We have checked and edited our manuscript carefully, we hope it is now easier to follow.

REVIEWERS' COMMENTS:

Reviewer #1 (Remarks to the Author):

Thank you for addressing my concerns, the manuscript is much improved!